# Long-term water stress and drought assessment of Mediterranean oak savanna vegetation using thermal remote sensing

María P. González-Dugo[1*], Xuelong Chen[2,3], Ana Andreu[1], Elisabet Carpintero[1], Pedro J. Gómez-Giraldez[1], Arnaud Carrara[4], Zhongbo Su[5]

[1]IFAPA, Consejería de Agricultura, Pesca y Desarrollo Rural. Apdo. 3048 ES-14071 Cordoba, Spain

[2] Key Laboratory of Tibetan Environment Changes and Land Surface Processes, Institute of Tibetan Plateau Research, Chinese Academy of Sciences, Beijing, China

[3] CAS Center for Excellence in Tibetan Plateau Earth Sciences, Chinese Academy of Sciences, Beijing, China

[4] Fundación CEAM. Parque Tecnológico,14 Calle Charles Darwin. Paterna (Valencia), Spain 46980

[5]Faculty of Geo-Information Science and Earth Observation, University of Twente, Enschede, the Netherlands

*Correspondence to*: María P. González-Dugo (mariap.gonzalez.d@juntadeandalucia.es)

**Abstract.** Drought is a devastating natural hazard that is difficult to define, detect and quantify. The increased availability of both meteorological and remotely sensed data provides an opportunity to develop new methods to identify drought

conditions and characterize how it changes over space and time. In this paper, we applied the surface energy balance model SEBS (Surface Energy Balance System) for the period 2001-2018, to estimate evapotranspiration and other energy fluxes over the *dehesa* area of the Iberian Peninsula, with a monthly temporal resolution and 0.05º pixel size. A satisfactory agreement was found between the fluxes modelled and the measurements obtained for three years by two flux towers located over representative sites (RMSD = 21 W m$^{-2}$ and R$^2$ = 0.76, on average for all energy fluxes and both sites). The estimations of the

convective fluxes (LE and H) showed higher deviations, with RMSD = 26 W m$^{-2}$ on average, than Rn and G, with RMSD = 15 W m$^{-2}$. At both sites, annual ET was very close to total precipitation with the exception of a few wet years in which intense precipitation events that produced high run-off, were observed. The analysis of the anomalies of the ratio of evapotranspiration (ET) to reference ET (ET$_o$) was used as an indicator of agricultural drought on monthly and annual scales. Hydrological years 2004/2005 and 2011/2012 stood out for their negative values. The first one was the most severe of the series, with the highest

impact observed on vegetation coverage and grain production. On a monthly scale, this event was also the longest and most intense, with peak negative values in January-February and April-May of 2005, explaining its great impact on cereal production (up to 45% reduction). During the drier events, the changes in the grasslands and oak trees ground cover allowed a separate analysis of the strategies adopted by the two strata to cope with water stress. These results indicate that the drought events characterized for the period did not cause any permanent damage to the vegetation of *dehesa* systems. The approach tested has

proven useful for providing insight into the characteristics of drought events over this ecosystem and will be helpful to identify areas of interest for future studies at finer resolutions.

## 1 Introduction

Drought, which is a devastating natural hazard and globally widespread, has complex consequences across spatio-temporal
scales and sectors. Unlike other disasters, it is still a challenge to define, detect and quantify droughts (Sheffield and Wood, 2011), impeding most prevention and mitigation actions. When droughts affect savannas, the two canopies of this ecosystem, grasslands and trees/shrubs, suffer from different stresses: (i) the pasture production is reduced or lost, with a direct economic consequence resulting from the need to supplement animal feeding and, in more severe situations, the death or premature sale of animals; (ii) the decline and dieback of trees affect the ecosystem structure, jeopardizing the long-term conservation of the
system (Fenshan and Holman, 1999). Traditional agropastoral systems in arid and semiarid areas have developed strategies to cope with drought, such as diversifying crops and livestock, adding different animal species and breeds, or fluctuating herd sizes (Hazell et al., 2001). More recently, insurance services have started to offer insurance for damage to pasture production caused by water stress, providing farmers with a means to recover after a disaster. However, the slow onset of drought, the large extension of savanna areas, and their complex canopy structure, introduce additional difficulties to the challenge of
monitoring drought and assessing its adverse effects.

The increasing availability of global meteorological data and new remote sensing products, with advanced processing services and free and open data, offers an opportunity to characterize drought objectively, and to extend its analysis in space and time. Many indicators of drought using remote-sensing inputs have been developed in the last decades (Wardlow et al., 2012). Surface energy balance models (SEBM) provide a physically based rationale to combine the most often used remote-sensing
retrievals for drought monitoring: vegetation indices (VIs) and land surface temperature (LST). The VIs provide information about the amount and condition of the vegetation (Jackson and Huete, 1991), while the land surface temperature describes the state of the surface and the partitioning of the available energy into sensible heat (H) and latent heat (LE) or evapotranspiration (ET) (Kustas and Norman, 1996). SEBM have been used to provide ET estimations over agriculture (Anderson et al., 2015; Allen et al., 2011; Cammalleri et al., 2012; Andreu et al., 2015; Gonzalez-Dugo et al., 2009, 2012) and agroforestry systems
(Andreu, 2018a,b; Guzinski et al. 2018; Carpintero et al. 2016). In particular, the SEBS (Surface Energy Balance System) model (Su, 2002) presents a good compromise between the detailed parameterization of the turbulent heat fluxes for different states of the land surface, and the minimization of the input requirements of the model without the need of local calibration. The evapotranspiration of a canopy is a suitable indicator of its water status and a good measurement of the impact of water shortage on vegetation and the functioning of the ecosystem. Evapotranspiration and soil moisture anomalies have been widely
used for spatially distributed monitoring of agricultural drought (Anderson et al., 2016; Cammalleri el al., 2015; Sheffield et

al., 2004). These anomalies underline the abnormally dry conditions when compared to the usual state of an ecosystem, derived from historical data. Evapotranspiration anomalies were used here to assess drought and vegetation water stress in the holm oak savanna area of the Iberian Peninsula over a period of seventeen years.

The Mediterranean oak savanna, called *dehesa* in Spain and *montado* in Portugal, is the most extensive and representative agroforestry system in Europe, with more than 3 million hectares in the Iberian Peninsula (Moreno and Pulido, 2009). It is a man-made ecosystem that maintains a fragile balance between its multiple uses (livestock, cereal crops, cork, hunting, etc.) and the conservation of its natural resources. The *dehesa's* diversity of habitats, giving refuge to a large number of species (Díaz et al., 1997), is especially recognized, and it is listed as having community-wide interest in the EU habitat directive (92/43/EEC). It is a water-controlled system, with its productivity directly dependent on water availability. Mediterranean oaks can minimize the effects of water scarcity through a combination of physiological mechanisms that occur over a range of time scales (Rambal, 1993). However, an additional problem to the recurrent water scarcity, is the identification of low soil water content as an initiating factor involved in the severe oak decline affecting a large area of *dehesa* since the early 1980s (Sánchez et al., 2002). Drought events impede the growth of *Quercus ilex* seedlings and increase their susceptibility to *Phytophthora cinnamomic* (Corcobado et al., 2014), the main biotic factor responsible for this decline (Sánchez et al., 2002).

Similarly to other savanna ecosystems, the different components of *dehesa* structure (sparse tall vegetation, large areas of grasses, shrubs, and bare soil), contribute differently to the turbulent exchange and radiative transfer, hindering its modeling, especially when compared with more homogeneous landscapes. In addition, these vegetation layers differ in phenology, physiology and function: while most trees are evergreen and have access to deep sources of water all year, the herbaceous layer only taps water from the first centimeters of soil and dries up during summer. The combination of the different functioning and characteristics of the system components affects the exchange of sensible and latent heat flux, resulting in a high spatial and temporal flux variability difficult to account for in model parametrization and algorithms. This structure appears to play an important role in savannas' resilience, making the system an efficient convector of sensible heat and keeping the canopy surface temperature inside the adequate range for survival (Baldocchi et al., 2004).

In this work, a surface energy balance model, SEBS (Surface Energy Balance System) (Chen et al., 2013, Su, 2002) has been applied to estimate evapotranspiration and other energy fluxes from 2001 to 2018 over the *dehesa* areas of Spain and Portugal. A first objective was to validate the energy fluxes produced by this model over the *dehesa* landscape. The second was to analyze the anomalies of the ratio of ET to reference ET as an indicator of agricultural drought in this environment at monthly and annual scale and use it to characterize the main drought events occurring in this period in space and time.

## 2 Data and methodology

The study was conducted over the oak savanna area of the Iberian Peninsula (Figure 1) using data from January 2001 to August 2018. This ecosystem covered 3.12 million ha in 2006 according to the European CORINE Land Cover inventory (CLC2006.

100 m - version 12/2009 https://www.eea.europa.eu/data-and-maps/data/ clc-2006-raster-4). The area has remained fairly stable during the study period, with changes of less than 1.5% between CLC2006 and the previous and posterior inventories, in 2000 and 2012.

## 2.1 SEBS model description

A revised version of the surface energy balance system model known as SEBS (Su, 2002) was used to estimate land heat fluxes integrating remote sensing and meteorological forcing data. A brief description of the model is presented below (for further discussion, see Su, 2002 and Chen et al., 2013). The latent heat flux (LE) was computed as a residual of the surface energy balance equation:

$$LE = \text{Rn} - \text{G} - \text{H} , \tag{1}$$

where Rn is the net radiation, G is the soil heat flux and H is the turbulent sensible heat flux. The net radiation is calculated using the following equation:

$$R_n = (1 - \alpha)SW_d + \varepsilon LW_d - \varepsilon \sigma LST^4 \tag{2}$$

Where $\alpha$ is broadband albedo; $SW_d$ is the downward short-wave radiation; $LW_d$, the downward long-wave radiation; $\varepsilon$, the land surface emissivity; $\sigma$, the Stefan-Bolzmann constant; and $LST$, the land surface temperature.

The soil heat flux is derived from its ratio to the net radiation ($\Gamma$) using equation 3:

$$G = R_n[\Gamma_C + (1 - f_C)(\Gamma_S - \Gamma_C)] \tag{3}$$

This ratio is assumed to be equal to 0.05 (Monteith, 1973) for surfaces with fully covered vegetation ($\Gamma_C$) and 0.315 for bare soils ($\Gamma_S$) (Kustas and Daughtry, 1990). The green canopy cover, $f_c$, is determined using the normalized difference vegetation index (NDVI) in equation 8.

Using equations 1 to 3 and energy balance considerations for limiting cases, the following reductions can be applied: (i) under the dry limit (equation 4), the evapotranspiration, $\lambda E_{dry}$ is assumed to become zero due to the limitation of soil moisture and the sensible heat flux, $H_{dry}$, is at its maximum,

$$\lambda E_{dry} = R_n - G - H_{dry} \equiv 0 \tag{4}$$

(ii) under the wet limit (equations 5 and 6), the evaporation takes place at potential rate, $\lambda E_{wet}$, only limited by the available energy at the given surface and atmospheric conditions. The sensible heat takes its minimum value, $H_{wet}$, with the internal resistance of the Penman-Monteith combination equation in the form written by Menenti (1984), $r_i \equiv 0$, by definition.

$$\lambda E_{wet} = R_n - G - H_{wet} \tag{5}$$

$$H_{wet} = \left( (R_n - G) - \frac{\rho C_p}{r_{ew}} \cdot \frac{e_s - e}{\gamma} \right) \Big/ \left( 1 + \frac{\Delta}{\gamma} \right) \tag{6}$$

where $\rho$ is the density of air; $C_p$ the specific heat at constant pressure; $e$ and $e_s$ are actual and saturation vapor pressure respectively; $\gamma$ is the psychrometric constant, $\Delta$ is the rate of change of saturation vapor pressure with temperature and $r_{ew}$ is the external or aerodynamic resistance. The sensible heat is computed according to the Monin-Obukhov similarity theory and limited by the dry and wet conditions. A complete description of the model and the use of the dry and wet limits can be found in Su (2002).

## 2.2 Model parametrization and dataset preparation

For the application of SEBS over *dehesa* area two surface variables, $f_c$ and the height of the canopy ($h_c$), have been adapted to the specific characteristics of this ecosystem. The green canopy cover ($f_c$) and leaf area index ($L$) were calculated using the following equations (adapted from Choudhury et al., 1994):

$$f_c = 1 - \left( \frac{NDVI_{max} - NDVI}{NDVI_{max} - NDVI_{min}} \right)^{\frac{1}{\xi}} \tag{7}$$

$L = -\frac{1}{k} \ln(1 - f_c)$     (8)

where $NDVI_{max}$ and $NDVI_{min}$, represent a surface fully covered by vegetation ($\sim$0.94) and completely bare ($\sim$0.15), respectively. The parameter $\xi$ represents the ratio of the canopy extinction coefficient ($K'$) to a leaf angle distribution term ($k$). $k$ was assumed to be equal to 0.5 for a random distribution of leaves, as the ecosystem contains erectophile grasses and planophile oak tree leaves (Andreu et al., 2019). $K'$ adopted a value of 0.8 obtained from experimental data and within the range proposed for NDVI by Baret and Guyot (1991). NDVI data was provided by MODIS instrument, averaging the 16-day original product to monthly scale.

The height of the canopy was computed to account for variations in the tree component. This variable is needed for calculating the momentum roughness length and thus, important for the sensible heat calculation. The tree stratum of the *dehesa* is quite homogeneous in composition, dominated by mature *Quercus ilex sp.*, and grassland canopy has a very high variability of low height herbaceous species. Considering these reasons, the ecosystem structure has been simplified to compute $h_c$ in the following way: A constant height of 8 m has been assigned to oak trees, which is multiplied by its ground coverage in each pixel. Oak $fc$ is computed annually using summer NDVI in eq. 7. During the summer, the grasslands are dry, and the only photosynthetically active vegetation contributing to the NDVI signal are the oak trees. The grassland height is low (< 1 m), affecting the effective canopy height of each pixel less than the trees, and it is also difficult to compute based on monthly vegetation indices given the high species variability. For this reason, the grassland height has been discarded and only the contribution of trees was considered to compute $h_c$. Thus, a single $h_c$ value was used for every month of a year. This

simplification of a complex system certainly may contribute to the error of modelled fluxes. However, it was an operative solution considering the scale of this study.

SEBS model was originally designed for instantaneous applications. Monthly calculations using the same model were demonstrated by Chen et al. (2014). The structure of the model was not changed, and the implementation differed in the input datasets. The model was applied over the entire Iberian Peninsula with a spatial resolution of 0.05° and a monthly input dataset. Satellite and meteorological input datasets are described in Table 1. All datasets were spatially averaged or subdivided to a common resolution of 0.05°.

The land surface temperature (LST) was provided by MODIS instrument, using the monthly mean of day and night LST product, which provides the most complete coverage. The accuracy of this product, a key variable in SEB models, was evaluated by Chen et al. (2017), supporting its applicability for climate studies and numerical model evaluation.

Meteorological data were provided by the ERA-Interim, a global atmospheric reanalysis data set from the European Centre for Medium-range Weather Forecast (ECMWF). Monthly means of daily means were produced by ECMWF as the average of the four main synoptic monthly means at 00, 06, 12, and 18 UTC. The forecast model, data assimilation method, and input datasets used to produce ERA-Interim can be found in Dee et al., (2011) and a description of the product archive in Berrisford et al., (2011).

To analyze model results, the monthly rainfall gridded data of the Climatic Research Unit (CRU) Time-Series (TS) Version 3.21 (Harris et al., 2014), provided by the Global Climate Monitor System (Camarillo-Naranjo et al., 2019), have been averaged over the *dehesa* area of the Iberian Peninsula.

## 2.3 Validation sites and model evaluation

Two experimental sites (Figure 1) with similar flux measurement instrumentation have been used to validate the evapotranspiration and other energy fluxes estimated using the SEBS model. Both eddy covariance towers, named Sta.Clo (Santa Clotilde, Andalusia, 38°12′N; 4°17′W, 736 m a.s.l.) and ES-LMa (Boyal de Majadas del Tiétar, Extremadura, 39°56′N; 5°46′W, 260 m a.s.l.) are located over *dehesa*-type ecosystems under similar management and a landscape of scattered oak trees with a fractional cover of around 20%, in southern and southwestern Spain, respectively. The convective fluxes of the systems are measured above the tree height (at 17 m in Sta.Clo and 15 m in ES-LMa) with closure balance errors of 20% and 14%, both values being within the range found by other authors (Foken, 2008; Franssen et al., 2010). For ES_LMa the processing of the data corresponded to the procedure standardized by Fluxnet network ((https://fluxnet.org/). For Sta.Clo, detailed information on the measurements and the processing of the data can be found in Andreu et al. (2018a and b). In this case, the comparison period was selected attending to the quality of the data and some months (3 of 36) were discarded due to missing information. Soil moisture, precipitation and other complementary measurements of the vegetation (reflectance, L, green canopy cover) were used to characterize the dynamics of the vegetation and the soil water status throughout the year.

The area contributing most to the fluxes measured was estimated by using Schuepp et al. (1990) and varied between 1 and 2 km. These footprints are lower than the pixel size of 5 km used for the application of the SEBS model. However, the homogeneity of the system, with similar tree ground cover fraction and pasture management at several kilometers around the towers supported the capacity of these sites to serve as a reference for the validation of modelled fluxes. In both cases, the good correspondence between the model input meteorological data at the tower's location and the ground measurements was verified (data not shown).

Monthly rainfall data for the seventeen years of the study was provided by the closest weather station to each site, located at 3 km and 16 km of Sta.Clo and ES-LMa towers, respectively. Both of them are operated by the Spanish Meteorology Agency (AEMET).

Model performance was quantified via the root mean square difference (RMSD) and the coefficient of determination ($R^2$) between the modeled and observed fluxes. In addition, the mean bias error (MBE), computed by taking the difference between predicted and observed, was used to assess model under- and over-estimations.

## 2.3 Water stress calculations

The relative evapotranspiration is the ratio of actual to potential or reference ET ($ET/ET_o$). It has been used as an indicator of crop water stress (Anderson et al., 2015, 2016), of drought (Anderson et al., 2011), and as a proxy for soil moisture (Su et al., 2003). The same approach is used worldwide in irrigation engineering to compute crop water requirements following FAO (24 and 56) guidelines (Doorenbos and Pruitt, 1977; Allen et al, 1998). The reason to normalize by $ET_o$ is to separate the ET signal component responding to soil moisture from variations due to the available energy. Anderson et al., (2011) showed that anomalies in $ET/ET_o$ were more strongly correlated with other drought indices as were anomalies in ET for most US climatic divisions, showing strong agreements in the southwest of the country, with a similar climate than the study area. The comparison of both variables anomalies has been also performed here.

Anomalous water stress conditions indicating drought were assessed here with the standardized values of relative ET. FAO56 reference ET (Allen et al., 1998) was selected to estimate the atmospheric evaporative demand (AED), given the difficulties of reproducing the biological control of the transpiration, even at potential rates, of the different types of vegetation conforming this ecosystem.

The vegetation water stress caused by the long dry summers of the Mediterranean climate can be considered to be the 'normal' state of the system for several months of the year. To identify unusually dry conditions indicating drought, standard (z) scores of this variable ($ET/ET_o$) for a given month/year have been computed. This standardization procedure assumes that the data follow a normal distribution. Some authors (Sheffield et al., 2004; Cammalleri et al., 2015) have pointed out that soil moisture and the water deficit index derived from it are generally characterized by a skewed distribution and can be statistically better represented using the beta distribution. In this case, the analysis of ET and relative ET monthly histograms (shown in the supplement) indicated that most months presented an approximately symmetric distribution, with skewness between -0.5 and

210 0.5 for both variables. Three months were moderately skewed and only one month (for ET) and two months (for ET/ETo) were slightly above one, backing up the use of z scores for the standardization of this variable. Annual drought analyses were performed by averaging monthly anomalies.

Drought intensity is defined here in terms of the maximum negative anomaly of relative ET values reached during an event (thus using the standard deviation as a measure of its departure from the mean) and the drought event duration as the successive
number of months with negative anomalies. To classify the events occurred during the study period, the following thresholds have been used: severe drought (anomalies <=-1.5); moderate drought (anomalies between -1 and -1.5) and mild drought (anomalies between -1 and 0). These classes are used for both annual and monthly time steps.

Two variables, vegetation coverage ($f_c$) and rain-fed wheat production, have been selected as drought impact indicators. The vegetation condition and the failure of crops are known consequences of a declining soil moisture and both have been used
previously as indicators of drought (Liu and Kogan, 1996; FAO, 1983). Winter cereals are the main cropping system of these areas, in which the low fertility of the soils does not allow a more intense agricultural use. Its growth cycle is similar to that of the natural grasslands, with both of them escaping drought and coping with the long summer dry season by completing its life cycle before serious soil and plant water deficits develop. Given that no irrigation is provided, the impact of moisture deficits over its yield can be consider an indirect indicator of the impact of drought on *dehesa* herbaceous vegetation. Annual yield
statistics (http://www.mapama.gob.es/es/estadistica/temas/estadisticas-agrarias/agricultura/esyrce/) have been gathered and aggregated for the *dehesa* area (Figure 1).

## 3 Results and discussion

### 3.1 Model validation

The comparison of SEBS model estimation of monthly energy fluxes with measurements at the two EC towers during a total
230 of six years, 2009 to 2011 for ES-LMa and 2015 to 2017 for Sta.Clo, displayed in Figure 2, generally showed good agreement, with an average root mean square difference (RMSD) of 21 W m$^{-2}$ and R$^2$ of 0.76, for all energy fluxes and both sites. The estimations of the convective fluxes (LE and H) show higher deviations, with RMSD = 26 W m$^{-2}$ on average, than Rn and G, with RMSD = 15 W m$^{-2}$. Model performance at ES-LMa site was, in general, superior to that at Sta.Clo, with all the statistics metrics computed for the comparison (RMSD, MBE and R$^2$) presenting lesser dispersion and slightly lower errors. LE was
235 slightly overestimated at both sites (MBE =10.3 and 2.8 W m$^{-2}$ at Sta.Clo and ES-LMa, respectively), which is in agreement with previous applications of the model (Michel et al., 2016). This overestimation was particularly significant for some springtime months at Sta.Clo, when the sensible heat was underestimated by the SEBS model (Chen et al., 2019). It is worth noting than the model forces the closure of the energy balance, and the error in LE can be attributed to the propagation of errors in all the other balance components. However, LE estimations presented a similar or lower RMSD than other
applications of the SEBS model (Chen et al. 2014; Vinukollu et al, 2011). In particular, the work by Chen et al. (2014) estimated

energy fluxes over China at the same temporal scale and with similar input databases. The comparison with measurements at 11 Chinese flux towers presented results that were very close to the ones obtained by this application. Mean RMSDs for all fluxes were alike (RMSD = 22 W m$^{-2}$ was reported by Chen et al. (2014)), with a marginally better performance for convective fluxes and a poorer one for Rn and G (RMSDs in China were 22 and 24 W m$^{-2}$ for convective fluxes and, Rn and G, respectively).

Figure 3 presents the evolution of modelled ET and ET$_o$, ET/ET$_o$ and measured precipitation from 2001 to 2018, aggregating the hydrological year (between October 1$^{st}$ and September 30$^{th}$) at the two experimental sites. It can be observed that annual ET variations for the period followed a similar pattern of precipitation at both sites, confirming the predominant control of water availability over the evaporation in these systems. This control is consequently extended to ecosystem productivity and in most years the water consumption, coupled to biomass production, is close to the total rainfall. Tree density is similar at both sites and the differences in water consumption between them are explained by variations in annual pasture production, due to differences in water availability and soil properties. Very wet years, and those with average rainfall but intense precipitation events producing an increase in run-off, did not follow this pattern. This can be observed by the run-off recorded at Sta.Clo watershed reservoir (Figure 3a). The main land-use of this small watershed (48.4 km$^2$) is *dehesa*, but other uses can be found as well, such as olive orchards and field crops.

Annual run-off measurements followed a close relationship (data shown in the supplement, Figure S2) with the annual aridity index (Budyko, 1974) estimated at Sta.Clo following Arora (2002), as the ratio between potential evaporation and annual precipitation. On average, we found aridity indices of above one at both sites, indicating dry regions where the evaporative demand cannot be met by precipitation. In this case, AED was computed using Penman-Monteith for comparison purposes. Sta.Clo site is noticeably less arid than ES-LMa, with an aridity index equal to 2.9 and 3.75 on average for the 17 hydrological years at Sta.Clo and ES-LMa, respectively, with both of them falling under the category of a semi-arid climate regime (Ponce et al., 2000). The two sites presented similar annual ET$_o$ values for the period (Figure 3), but annual precipitation was around 200 mm higher, on average, at Sta.Clo, with a higher and more variable ET/ET$_o$ throughout the years. What can also be observed in Figure 3 is the complementary relationship between actual and reference evapotranspiration at this temporal scale, with the sum of annual ET and ET$_o$ approaching a constant value at both sites, confirming the complementary hypothesis (Bouchet, 1963; Morton, 1975; Brutsaert and Stricker, 1979).

**3.2 Annual drought monitoring and impact assessment**

Drought was characterized on an annual scale over the experimental sites and the whole area of the *dehesa* of the Iberian Peninsula using the relative evaporation anomalies. Figure 4 presents their evolution for the two sites throughout the study period. A clear similarity can be observed in the main negative anomalies, which identify the most severe droughts during the years 2004/05 and 2011/12 at both sites, despite the differences in aridity and the distance (Figure 1) between them, indicating the extended area and intensity of both events. Differences are more evident in the case of the mild droughts, occurring at both

sites but with different intensities during two periods, 2007 to 2009 and 2016 to 2018.

When the whole *dehesa* area is considered (Figures 5 and 6), a more complete view of the general intensity, impact, and spatial
distribution of those dry periods, can be obtained. Figure 5 aggregates, for the total *dehesa* area, the evolution of the relative
ET anomalies, together with the exchanges of energy between the surface and the atmosphere, the green canopy cover, and
the production of rainfed wheat. The last two variables were selected as indicators of the impact of water scarcity on the system.

The two severely dry years identified at the experimental sites were the driest ones for the entire *dehesa* area, with 2004/2005
standing out as the most severe event of the time series. None of them lasted more than one year. For these two dry years, a
reduction in the latent heat can be observed when compared to the complete series, producing a swap with the sensible heat in
the second position in magnitude of the energy balance components. A rise in the surface temperature, increasing the difference
with the air temperature, is also observed for those dry years. The order of severity in dryness, established by the magnitude
of negative values of ET/ET$_o$ anomalies, is also observed in their impacts over the system (Figure 6). In 2004/05, the wheat
production in the area was reduced by almost half of the average (45%) for the period analysed, and the vegetation groundcover
fraction fell by 20% compared to the average of the same period. This severe drought affected the entire Iberian Peninsula,
with the Spanish and Portuguese cereal and hydroelectricity productions decreasing by 40% and 60% with respect to the
average (Garcia-Herrera et al., 2007) and a 10% reduction in total EU cereal yields (UNEP, 2006). The event during 2011/2012
was among the largest and most severe ones in Europe for the 18-year simulation period analysed by Cammalleri et al. (2015),
contributing to a global decline in grain production.

Figure 6 shows maps of ET/ET$_o$ anomalies in Iberia for the seventeen years of the study, highlighting the *dehesa* area of interest
in this work. The spatial variability of these anomalies for most years is significant, although prevalently dry and wet years
can be distinguished. In 2004/05 and 2011/12, the drought was severe and affected most of the area of interest, as the aggregated
values of Figure 5 also point out. In 2008/09, the water stress was milder in the western area, as can be observed in Figure 6,
as at the experimental site of Sta.Clo (Figure 4) located in this part of the region. The recovery of the vegetation water status,
in most areas, was achieved the year following dry ones.

### 3.3 Monthly drought analysis

The monthly evolution of relative evapotranspiration anomalies is displayed in Figure 7a, with negative values indicating water
stress conditions highlighted in red. Absolute ET and ET$_o$ values, used to calculate these anomalies, are shown in Figure 7b
together with monthly rainfall for the period. One can observe the alternation of complementary and parallel characteristics of
ET and ET$_o$ throughout the year. The longest complementary period indicating water-limited ET conditions, starting in May
for most of the years, is confirmed by the decreasing trend in rainfall starting in that month. At the end of the summer when
the first rains arrive, the trend of ET and ET$_o$ changes, producing a secondary peak in ET, much weaker than the one earlier in
the year, that lasts until the energy-limited parallel phase starts in November. Both variables follow a concurrent rise from

January until the soil water deficit limits ET again.

The annual fluctuations of the green canopy cover (thick green line in Figure 7a) followed the expected seasonality of Mediterranean vegetation, corresponding to the dynamics of ET and $ET_o$ changes. The maximum coverage (March and April) corresponds to the peak of grassland production (and ET although with different shape) and the minimum appears during the dry summer, only endured by the oak trees. In some years, the growing season presents a bimodal shape, with an initial peak produced by autumn pastures, which is also reflected in ET values. It can be observed mostly in wet years (e.g. 2003, 2007,

2011), with the vegetation growth following a pattern that can be related to the soil water availability, represented here by the $ET/ET_o$ anomalies.

The duration and intensity of each drought event help to explain the response of the vegetation during these periods. In this sense, the two main drought events identified on an annual scale (2004/05 and 2011/12), presented dryer than normal conditions during the whole or most of the year. The first event was longer (sixteen months in the first case, prolonging the

drought to the beginning of the following year) and with higher negative values than the second one, of an eleven-months duration, explaining the greater impacts detected on the vegetation and cereal yield. Other dry periods, in 2009, 2017 and 2018, presented consecutive negative anomalies for ten to eleven months but, in some cases, the non-homogeneous distribution of the drought observed in Figure 6, may have undermined the impact analysis on this aggregated spatial scale. In terms of impact assessment, the time of the year with peak negative anomalies is important, with springtime events producing greater impacts

(e.g. in 2004/2005 the highest negative values corresponded to January, February, April, and May of 2005).

During the dry years, the annual vegetation growth pattern varies with respect to the typical one, depending on the duration and severity of drought events. The dynamics of the vegetation in this system allows for a separate analysis of the effect of water scarcity over trees and pastures. The dashed green lines (Figure 7a) show the changes in annual maximum and minimum values of $f_c$, with the maximum ones mostly expressing the impact on pasture, and the changes in the minimum ones represent

only the impact over the tree canopy. The decreases in pasture $f_c$ are more pronounced than changes in oaks $f_c$, as grasslands are more abundant, and their roots are mostly located in the first centimetres of soil. On the contrary, the rooting system of the oak tree is in fact adapted to the regular dry periods of the Mediterranean climate, exploring a large volume of soil that can reach maximum values of around 5 m in depth and 30 m in horizontal extension (Moreno et al., 2005). The small decreases, observed in oaks $f_c$ in Figure 7a during dry years, generally recovered within one or two years. This response of the tree leaf

area is associated with low frequency oscillations, such as annual rainfall (Poole and Milles, 1981). This is also supported by the variance observed in $f_c$ that can be explained by the anomalies of relative evapotranspiration of previous months. During the spring, the highest correlation coefficients are obtained for the previous two or three months (e.g., average $f_c$ for the peak month, April, is correlated with average anomalies from February to April with an $R^2$ equal to 0.76 and with anomalies of the previous year with an $R^2 = 0.52$). However, during the summer, the coverage of the vegetation can be better explained by what

has happened during the previous year (e.g., $R^2$ is equal to 0.39 for average August $f_c$ and the anomalies of the two previous months, and 0.64 for the anomalies of the year), suggesting that those values of $f_c$ might be linked to processes occurring at

different time scales.

A more detailed analysis is required, but these results support the conclusion that the drought events characterized for this period did not cause any permanent damage to the vegetation, considering both the grasslands and the oak trees.

Similar results can be derived from the analysis of ET anomalies. Figure 8 presents a comparison of monthly anomalies of ET, ET/ET$_o$ and $f_c$. The anomalies of ET and ET/ET$_o$ showed a high similarity for the conditions of the study, with correlations of $R^2 = 0.76$ at monthly scale and $R^2 = 0.82$ at seasonal scale (results presented in supplement figures S3 and S4). It suggests that ET anomalies could be an option to monitor drought in *dehesa* areas. Nevertheless, the computation of ET$_o$ does not require additional variables than those already used by the energy balance models, with a quite straightforward computation. Once 345 actual ET is estimated, the computation of ET/ET$_o$ takes very little effort and adds some confidence to the focus on the soil moisture signal. Regarding the evaluation of $f_c$ anomalies, it can be derived that the drought events identified using this variable would have been the same as using ET or ET/ET$_o$, but with different intensities and duration. The main differences can be found during the cold winter months when the vegetation is largely dormant. In these cases, the anomalies of $f_c$, similarly to the performance of other indices based on vegetation as the Vegetation Condition Index (VCI) (Heim, 2002) have a limited 350 utility. The results are more comparable and could be more useful during the growing season.

## 4 Conclusions

The SEBS model was used to estimate monthly energy fluxes over the *dehesa* area of the Iberian Peninsula from January 2001 to August 2018. There was a satisfactory agreement between modelled fluxes and measurements obtained for three years over 355 two sites that are representative of the ecosystem.

At both sites annual ET was very close to total precipitation, with the exception of a few wet years and those in which intense precipitation events producing a high run-off were observed. Average aridity indices for the 17 hydrological years of 2.9 and 3.75 were computed at Sta.CLo and ES_LMa, respectively, indicating that their evaporative demand cannot be met by annual precipitation of these sites.

Drought has been characterized on an annual and monthly scale over the experimental sites and the whole area of *dehesa* of the Iberian Peninsula using relative evaporation anomalies (ET/ET$_o$). At the annual scale, the negative anomalies of two years, 2004/2005 and 2011/2012, stood out during the study period at the experimental sites and the entire *dehesa* area. However, a recovery of average values is observed in the years following the dry ones, indicating the absence of prolonged droughts for the period. Maps of ET/ET$_o$ anomalies showed that most of the *dehesa* area was affected in those dry years. These maps 365 complemented the averaged data, providing spatial information about regional impacts that could be useful for a more detailed

analysis.

On the monthly scale, the drought event of 2004/05 is confirmed as being the longest and the most intense event, with sixteen consecutive months of negative anomalies (from October 2004 to January 2006). Peak negative values in January-February and April-May of 2005 explain the important impact on cereal production. The dynamics of the vegetation strata on a monthly

scale allows for a separate assessment of water stress impacts on oaks and pastures. The different behaviour observed in vegetation ground cover during the drier events in months with a preponderant presence of grasslands, compared with months in which only oaks were active, is consistent with the different strategies adopted by the two strata to cope with water stress. In addition, the correlation of monthly vegetation fractional coverage with previous short or medium-term anomalies (from two months to one year) suggest that those values might be linked to processes occurring on a different time scale, depending

on whether the grassland or the tree is the predominant vegetation.

These results back up the conclusion that the drought events characterized for this period did not cause permanent damage to the vegetation of *dehesa* systems, considering both the grasslands and the oak trees. The approach proved useful for providing insights into the characteristics of drought events over this ecosystem, for defining and identifying areas of interest for future studies at finer resolutions.

**Code and data availability**

SEBS code is available in GitHub repository to download (https://github.com/TSEBS/SEBS_Spain). Validation data of ES-LMa site is available at the European Fluxes Database Cluster (http://www.europe-fluxdata.eu/home/site-details?id=ES-LMa) and data of Sta.Clo site may be distributed on request to the principal investigator of Sta. Clotilde experimental site (M. P. González-Dugo, IFAPA, mariap.gonzalez.d@juntadeandalucia.es).

**Acknowledgement**

This work has been funded by the OECD Cooperative Research Programme: Biological Resource Management for Sustainable Agricultural Systems (contract JA00084693) and the projects PP.PEI.IDF201601.16 and PP.PEI.IDF2019.004, 80% cofunded by the European Regional Development Fund, AOP 2014-2020. Additional support was provided by RTA2014-00063-C04-02 INIA-FEDER and PID2019-107693RR-C22 projects (MCIU/AEI/FEDER, UE). X.C. was supported by the National

Natural Science Foundation of China (41975009), and A.A. by EU Horizon 2020 Marie Skłodowska-Curie grant agreement No. 703978. We would like to thank the owners and workers of Santa Clotilde experimental site, as well as the group managing the experimental site of Las Majadas for the eddy covariance measurements and the additional data. We also thank the anonymous reviewers, whose comments have improved the manuscript.

## Author Contributions

M.P.G.-D. conceived the original idea, analyzed the data and took the lead in writing the manuscript; X.C. and Z.S. designed the model, the computational framework, and contributed to the interpretation of the data; M.P.G.-D. and X.C. collected the input data and performed the numerical calculations; A.A., E.C., P.G.G. and A.C. collected and analyzed the validation data and reviewed the paper. All authors provided critical feedback and helped to shape the manuscript.

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

**Table 1.** Input datasets used to calculate the surface energy fluxes over the Iberian Peninsula from 2000 to 2018

| Variable | Full variable name | Data source | Spatial resolution | Temporal resolution of input products | Method |
|---|---|---|---|---|---|
| $SW_d$ | downward surface shortwave radiation | ERA Interim(ECMWF)[*] | 0.7° | 1 month | Reanalysis |
| $LW_d$ | downward surface longwave radiation | ERA Interim(ECMWF) | 0.7° | 1 month | Reanalysis |
| $T_a$ | air temperature | ERA Interim(ECMWF) | 0.7° | 1 month | Reanalysis |
| Q | specific humidity | ERA Interim(ECMWF) | 0.7° | 1 month | Reanalysis |
| u | wind speed | ERA Interim(ECMWF) | 0.7° | 1 month | Reanalysis |
| P | surface pressure | ERA Interim(ECMWF) | 0.7° | 1 month | Reanalysis |
| LST | land surface temperature | MOD11C3 V5[**] | 0.05° | 1 month | Satellite |
| $\alpha$ | albedo | GlobAlbedo[***]/MODIS[**] | 0.1° | 1 month | Satellite |
| NDVI | Normalized Difference Vegetation Index | MOD13C1 V5/MYD13C1 V5[**] | 0.01° | 16 days | Satellite |
| fc | fractional canopy coverage | Derived from NDVI using eq.7 | 0.01° | 16 days | Satellite |
| L | leaf are index | Derived from fc using eq.8 | 0.01° | 16 days | Satellite |
| hc | canopy height | Derived annually from summer NDVI | 0.01° | 16 days | Satellite |

[*]http://apps.ecmwf.int/datasets/data/interim-land/type=fc/
[**]https://modis.gsfc.nasa.gov
[***]http://www.globalbedo.org/index.php

## Figure Captions

Figure 1: Distribution of oak savanna area in the Iberian Peninsula. Location of Sta.Clo (Santa Clotilde) and ES-LMa (Las Majadas) validation sites and pictures of both eddy covariance flux towers.

Figure 2. Comparison of monthly energy fluxes of latent heat (LE), sensible heat (H), net radiation (Rn) and soil heat flux (G) estimated using the SEBS model at a monthly scale and observed fluxes at each oak savanna site: ES-LMa (LA) for the years 2009-2011 and Sta.Clo (SC) for the years 2015-2017.

Figure 3. Evolution of annual rainfall, ET, ETo and ET/ETo at ES-LMa site (a) and Sta.Clo site (b), and annual run-off at Sta.Clo watershed from the hydrological years 2001/02 to 2017/2018.

Figure 4. Annual anomalies of relative evapotranspiration at ES-LMa and Sta.Clo experimental sites estimated using the SEBS model from 2001/02 to 2017/18.

Figure 5. Evolution from 2001/02 to 2017/18 of annual anomalies of relative evapotranspiration, energy balance components, air and surface temperature, vegetation ground fraction cover and rainfed wheat yield, aggregated for the whole oak savanna area of the Iberian Peninsula.

Figure 6. Spatial distribution of annual anomalies of relative evapotranspiration for the oak savanna area of the Iberian Peninsula from 2001/02 to 2017/18, the average ET/ETo for the period and its standard deviation (STD)

Figure 7. (a) Monthly evolution of evapotranspiration anomalies (blue line) of the oak savanna area of the Iberian Peninsula from January 2001 to August 2018, with negative values indicating drier than normal conditions (depicted in red), and green canopy cover (green line). The dashed green lines connect the annual maximum and minimum values of $fc$; (b) Monthly evolution of and rainfall, $ET_o$ and ET in the same region and time interval.

Figure 8. Comparison of monthly negative anomalies of ET, ET/ET$_o$ and $f_c$ for the entire oak savanna area of the Iberian Peninsula from January 2001 to August 2018.

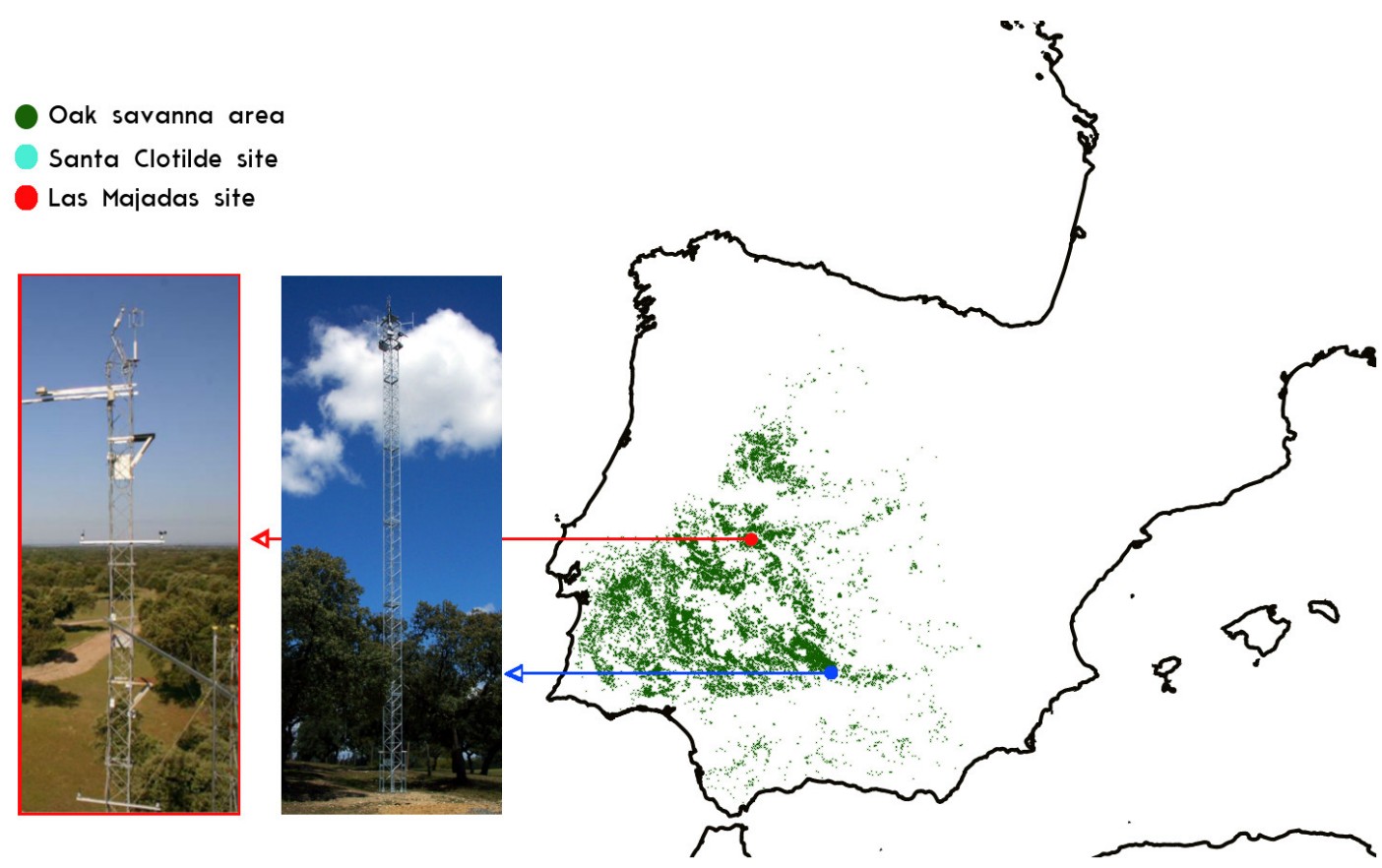

**Figure 1**

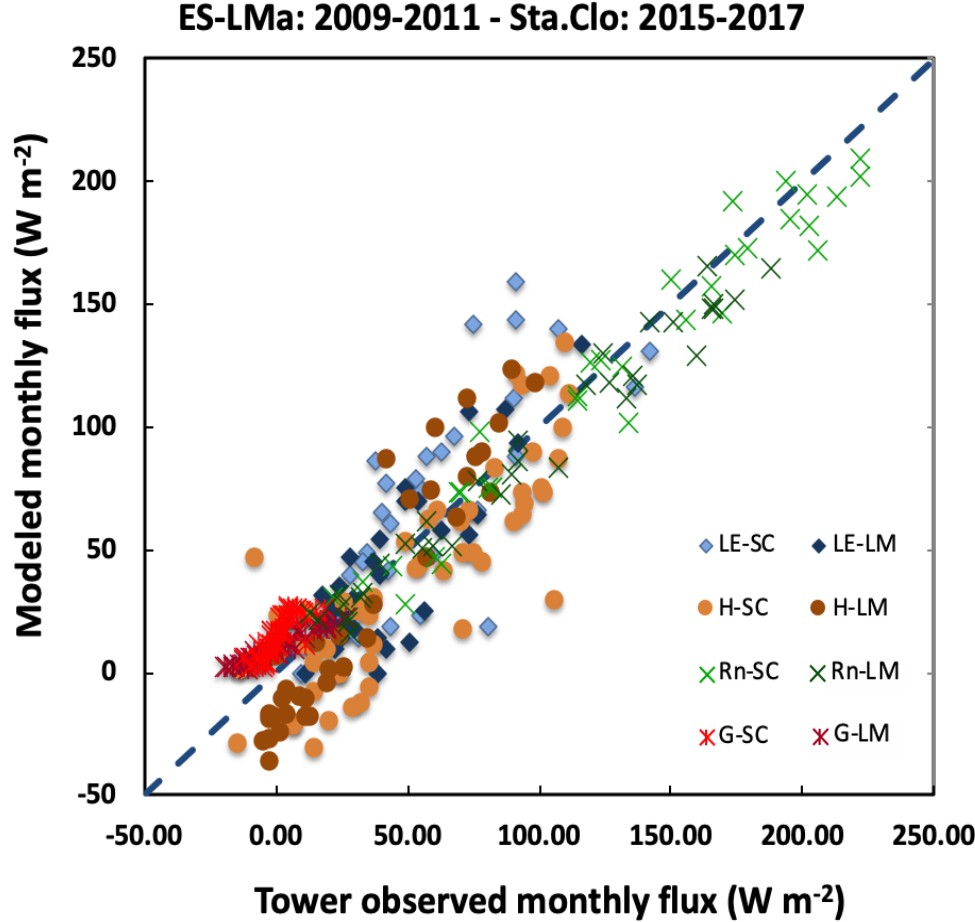

**Figure 2**

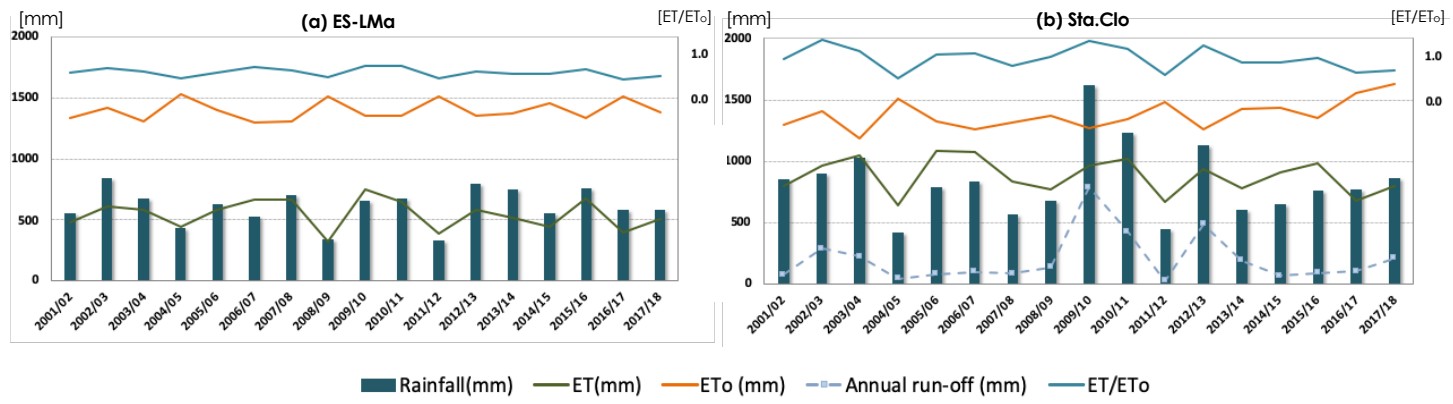

**Figure 3**

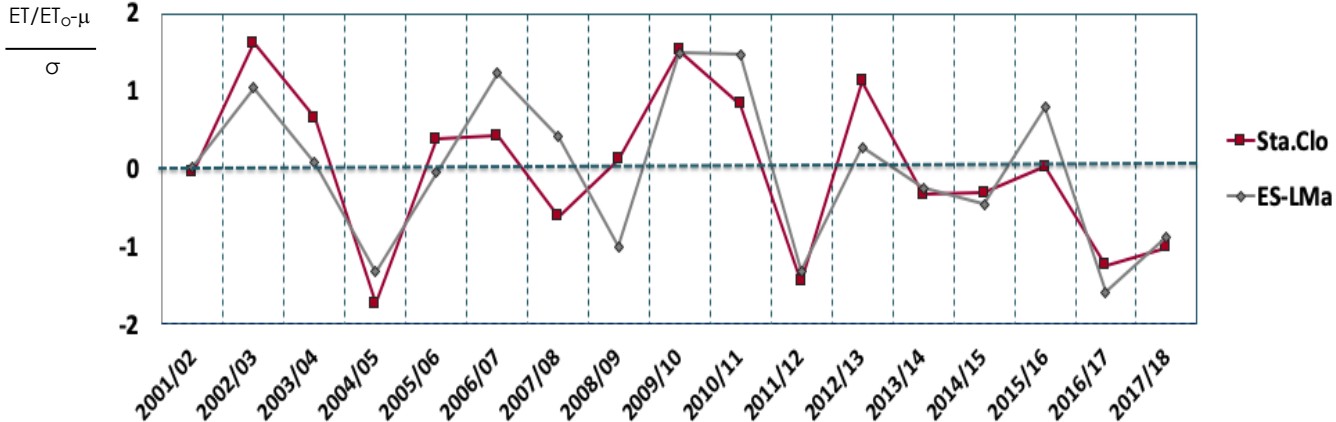

**Figure 4**

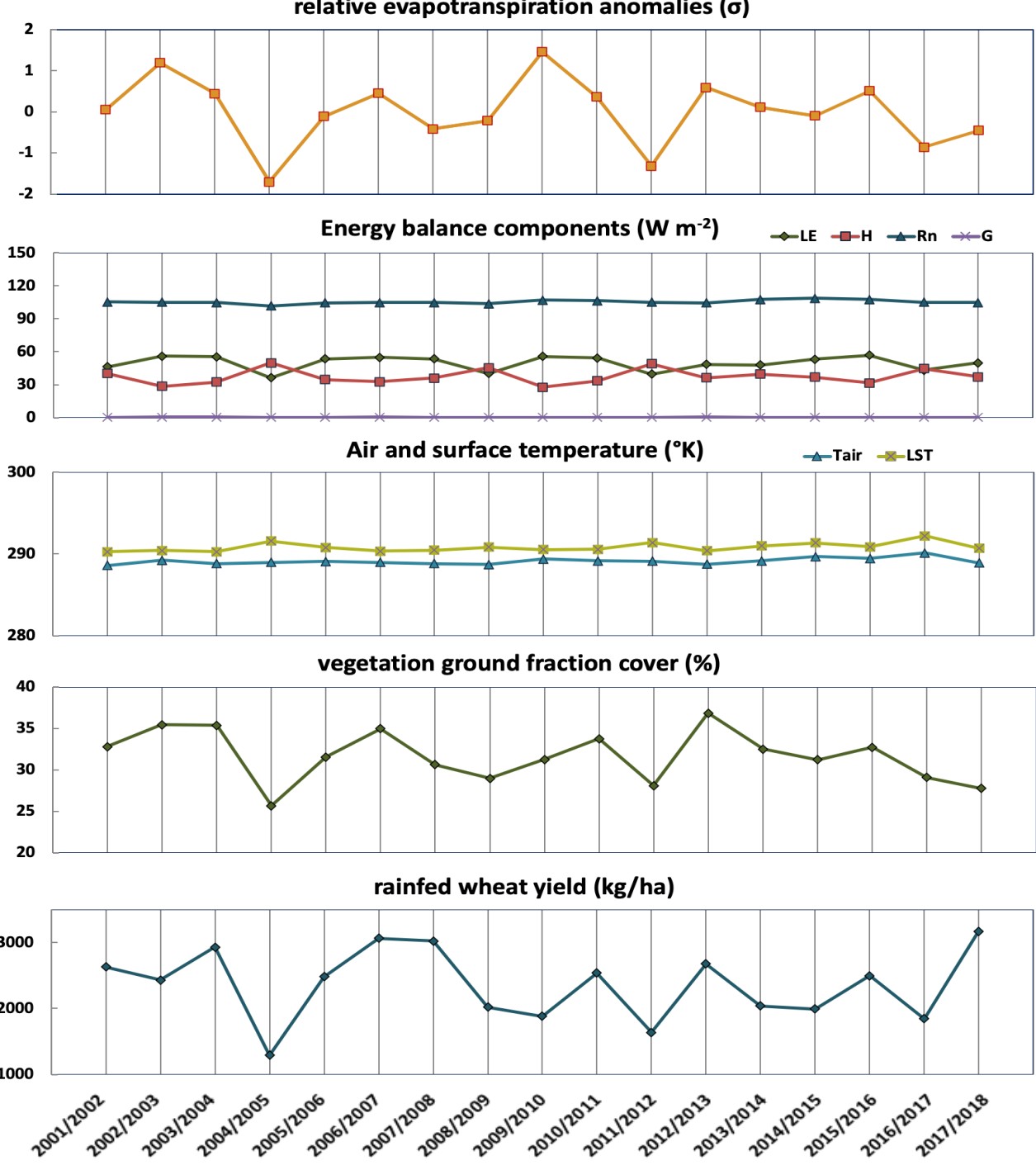

**modified Figure 5**

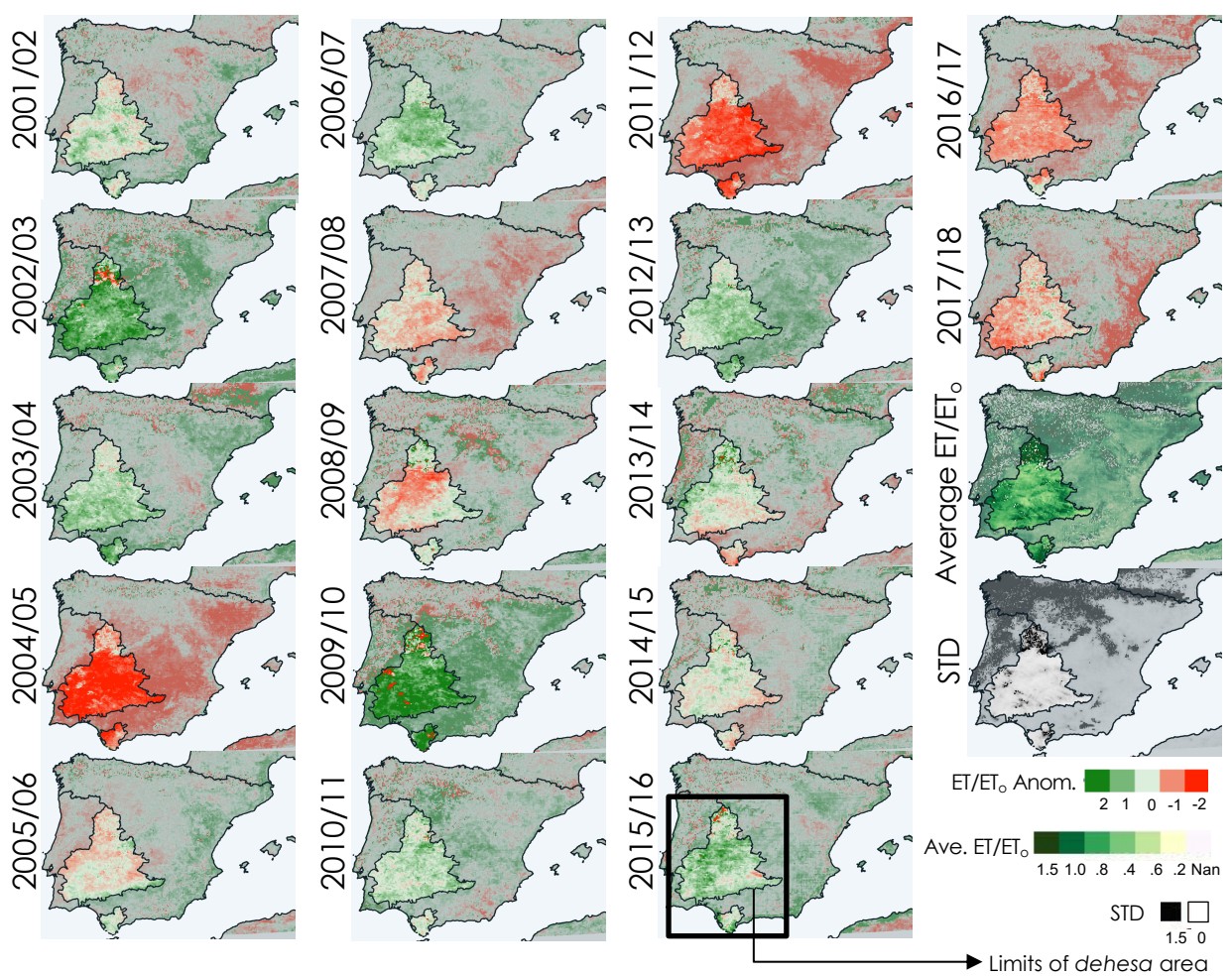

**modified Figure 6**

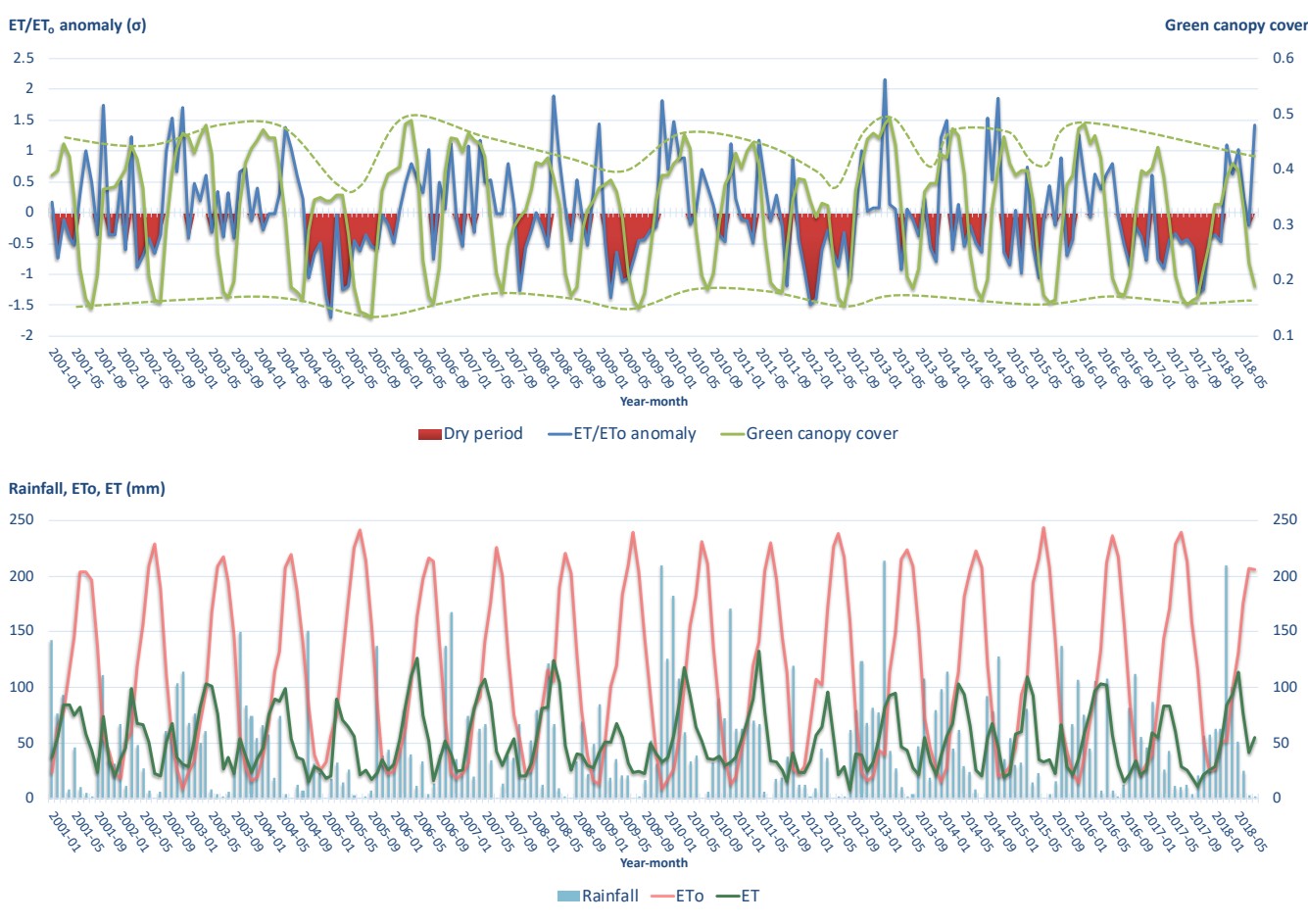

**Figure 7**

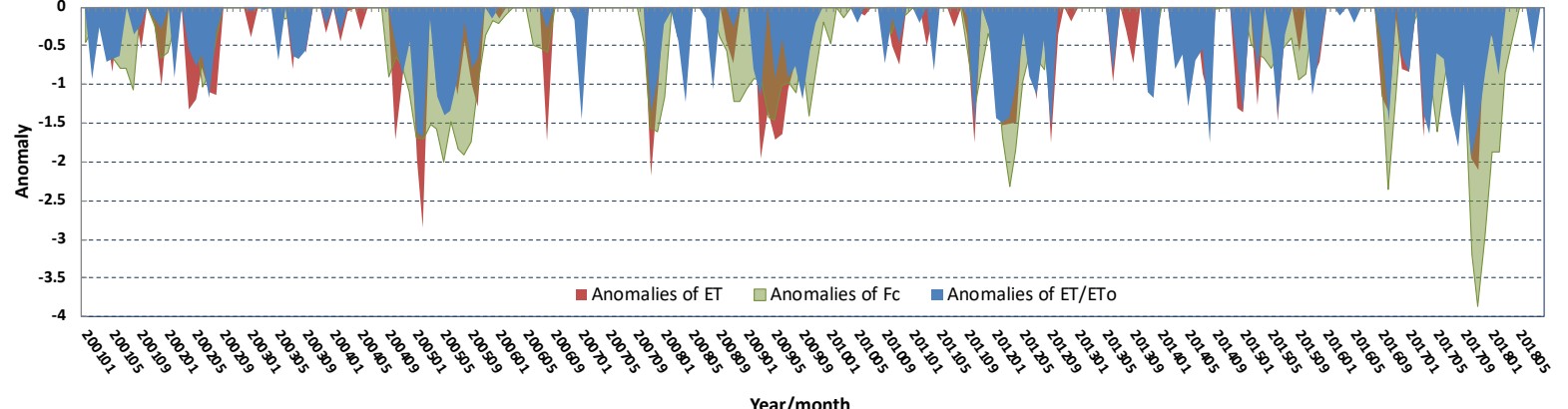

**new Figure 8**