# Peer review of "Long-term water stress and drought assessment of Mediterranean oak savanna vegetation using thermal remote sensing"

_Hydrology and Earth System Sciences, 2020_

## Referee Comment (RC1) · Anonymous Referee #1 · 12 Jun 2020

General Comments The manuscript present am interesting study using a long-term dataset to characterize the impact of water stress on the dehesa region of Spain. Overall, study was well designed, the paper is well written, and the results and conclusions are fully supported. however there are a few aspects of the study that need some clarification. The concerns, along with handful of minor grammar and typographical errors, are noted below.

Specific Comments 1. Line 13: The sentence beginning "Drought is a ..." might be expressed more clearly as : "Drought is a devastating natural hazard that is difficult to define, detect and quantify."

2. Line 13: The sentence beginning "Global meteorological data ..." is oddly constructed. It might be more clearly expressed as" The increased availability of both meteorological and remotely sensed data provides an opportunity to develop new methods to identify drought conditions and characterize how it changes over space and time."

3. Line 26: The sentence beginning "During the drier ..." is unclear and needs revision.

4. Line 34: The sentence beginning "Drought is a ..." could be expressed more clearly if constructed as: "Drought, which is both a devastating natural hazard and globally widespread, has complex consequences across spatiotemporal scales and sectors."

5. Line 43: Replace "slow-onset nature" with "slow onset".

6. Line 48: Indicators of what?

7. Line 53: The sentence beginning "LST and VIs" reads oddly. The authors seem to be saying that by combining information about the surface temperature and vegetation, remote sensing-based models can provide accurate estimates of ET. But, rather than statin that explicitly, the coach it in terms of vegetation indices etc.

8. Line 115: This paragraph is a bit unclear. The authors state the parameterization of green vegetation fraction and height are unique for the dehesa. Are the authors back calculating the leaf area index (L) using equations 8 & 9? If so, why? Also, there is no discussion of canopy height and how it's calculation is modified to better represent the dehesa.

9. Line 172: It would be helpful if the authors included a histogram and an estimate of the distribution skewness for ET and relative ET. From the description given here it appears quite small.

10. Line 182: Replace "presented a general good agreement" with "generally showed good agreement".

11. Line 184: Why the greater discrepancy for the turbulent fluxes compared to the

non-turbulent fluxes? Is this linked to imperfect closure for the flux measurements? Errors in partitioning the available energy between H and LE?

12. Line 206: The sentence beginning "Very low runoff ..." is redundant and could be omitted.

13. Line 207: Why isn't the relationship shown? Although it reasonable to suspect these two quantities would be correlated, a "close" relationship is a bit of a surprise. It would be useful to show this relational.

14. Line 207: Numerous metrics and indices have proposed been proposed over time to quantify aridity. It would be helpful to add a sentence or two to describe this index.

15. Line 222: Do the difference in the anomalies suggest local drought conditions? For example, during 2008/2009 there is a strongly negative value at the ES-LMa site while the value is slightly positive at StaClo. Would this indicate a local drought in the area about ES-LMa?

16. Line 253: it worth point out that the peak in the autumn is much weaker than the one earlier in the year.

17. Line 299: The phrase "and the more ..." also refers to ES-LMa, which was already discussed above. It appears to be an artifact from the writing process and should be deleted.

18. Figure 5: The word "fraction" is misspelled.

---

## Referee Comment (RC2) · Anonymous Referee #2 · 15 Jul 2020

General Comment

This paper deals with the modeling of drought in a oak savanna in Spain, where trees and pasture coexists, using ET estimates from thermal remote sensing data. I found the paper generally well written and well organized. The goal is clear, and the results sufficiently elaborate. However, I have three main concerns regarding the adopted methodology:

1) the SEBS model is well-known in the remote sensing community for "instantaneous" application at the satellite overpass time (eventually followed by upscaling procedures to daily/monthly scale). Here the model is used on monthly data, but the authors fail

to clarify how the model was adapted for the change in time scale (more details in the specific comment P8, L4).

2) The authors decided to use anomalies of the ratio ET/ETo as drought indicator. However, they do not provided neither evidences that this index perform better than others (e.g. even the simple ET), nor justification on why this index was used for the ecosystem under analysis (is it better suited for oak savanna than others?). Indeed, part of the study shift the focus on fc, because ET is not able to separate the behavior of trees and pastures. This analysis, even if interesting, is out of place give the declared goal of the study.

3) The authors used vegetation coverage and wheat productions as proxy of the drought impacts, without providing any justification for this choice. The first quantity is actually one of the input of SEBS, but is also weirdly used also for "validation", whereas the second is not necessarily related to drought impacts in a drought-resistant agro-pastoral system (see their words in P3, L6 of the manuscript).

In view of these considerations, I suggest the authors to revisit the manuscript to clarify these points before considering for publication. Some additional specific comments are also reported below, which I hope would be useful for improving the overall quality of the manuscript.

Specific comments

Title: I would replace the world "monitoring" with something else, since in my opinion monitoring implies something done in near-real time.

P2, L1: RMSD > xxx, and R2 < xxxx for all. . .

P2, L2-3: The details for each site are not needed in the abstract, especially after the previous sentence.

P2, L8: "with the first one being. . .". I suggest to move this to a new sentence.

P5, L4: Here I miss something that better links the previous description of the dehesa with the adopted modeling framework. In particular, why ET modeled by SEBS has been used? Is it a good option to capture the specificities of this environment (e.g. other options, such as dual source approaches, agri-forest modeling)?

P6, L3: I would suggest to write the eq. as LE = Rn - …. since you already introduced the concept of LE as residual.

P6, Eqs. (4) and (5). The second eq. is redundant.

P7, Rqs. (6) and (7). These two equations are confusing. In LEwet is computed via eq. (6), then Hwet needs to be defined in another way, or vice versa. Please clarify.

P7, L3. The way the limits are used needs a better clarification.

P7, L4. if Hwet is derived from Eq. (7), LEwet needs to be defined by an eq. that is not eq. (6) (e.g. Penman-Monteith as stated afterward).

P7, L5. "… a set of assumptions…". Please provide a brief description of these assumptions.

P7, L7. The role of canopy height is not clear at this point for a reader that is not familiar with the model. Please briefly introduce where and how hc plays a role. Also, the authors introduced a "revised version of the model… new bare soil resistance" (P5, L18), but the role of this new parameterization is not clear since there are no mention of resistance in the model description.

P8, L4. The SEBS model has been designed for "instantaneous" application at the time of LST acquisition. As a consequence, more details needs to be provided on how the authors adapted the model to work on monthly LST. I think that the idea is to use monthly LST as a "artificial" instantaneous LST for a theoretical average day, but some questions that needs to be addressed are: - how did you ensure consistency between the mosaicked monthly LST and 6h ECMWF meteo forcing? - How 16 days NDVI was used jointly with monthly LST? - How daily upscale was performed? - How monthly

upscale was performed?

P9, L1. Some details on the balance closure would be helpful. Was closure forced, and with which method? How were the data cumulated at monthly scale (I'm assuming some unavoidable missing data during the acquisition, any constrain on minimum data, etc.)?

P10, L13. It would be interesting to have a couple of words on the reason behind the use of ET/ETo rather than ET itself for the computation of anomalies. In my experience, there are many cases where ET anomalies are a better proxy of drought that ET/ETo ones. Ideally, the authors should add a test showing that ET/ETo outperform ET alone (especially with the latter being a more conservative approach, which does not need any additional quantity).

P10, L15. I have some issue with the use of fc as proxy of drought impacts, especially when fc is also one of the input of SEBS. If fc is a good proxy of drought impact, why we should use a complex model such as SEBS (which uses fc as input) to derive a quantity (ET) which performance is evaluated against fc. Why don't we use directly fc (or rather fc anomalies) at this point?

P10, L18. Similarly, I miss the connection between the impact of drought on the dehesa (a predominantly oak savanna) and wheat production. I know that having an independent estimate of drought impacts is tricky, but if the focus of the paper is specifically for the dehesa, you should justify better why wheat production is a good proxy of the drought impact on a likely drought-resistant, adapted oak savanna. The use of this quantity risks to lost the specificity of the work that you introduced earlier.

P11, L12. It would be better to have the results disaggregated fro seasons, in order to better highlight the impact of this seasonality in the error. This would help discussing the results, since drought may be mostly concentrate in some seasons. Also, since your goal is to use ET/ETo anomalies as proxy for drought, it would be much better to have in addition a validation of both ET/ETo values and z values against ground

data. Even if the length of the time series is quite short, it is important to show that the model is able to capture the year-to-year fluctuations, since this is what you want to reproduce. Often, ET estimates are "well" modeled only because the area has a strong yearly cycle.

P12, L1. It is weird to me that you show the yearly-aggregated data before the monthly one. Apart form that, Figure 3 is a good example of my consideration on P10, L13. Just looking at the plot, it seems that ET capture the same events that ETo if Precipitation is used as reference. What is the added value of using ET/ETo rather than ET alone?

P12, l15 to P13, L5. This whole paragraph seems a little out of topic to me. I suggest to reword to clarify the role in explaining drought in the region, or remove it completely.

P13, L13. Please define a mild drought. Also, it is not clear to me what is the role of this intercomparison between the modeled data over the two sites. Please clarify the aim of this comparison and justify the inclusion of a dedicated figure.

Fig. 5. Again, what is the added value of ET/ETo anomalies over ET alone (or, even worse, fc)? If anything, these figures are convincing me even more that a complex modeling framework is not needed, at least at annual scale. I'm sure that there is something more, but this is not discussed and justified by the accompanying text.

Fig. 6. There is an odd strikingly resemblance between the spatial patterns in the years 2004/2005 and 2011/2012. Can you elaborate a little more on that?

P15, L2. Is this the average over the whole dehesa? A single point? Other? Please clarify. Also, in case of the average, it would be interesting to see if also the spatial variability (std.dev) shows interesting results.

P15, L12. What about the intra-annual fluctuactions? Are they similar to ET/ETo z values also at this temporal scale? Any temporal delay?

P15, L19. Similarly to comment P13, L13, duration and intensity of drought needs to be defined in the methodology section.

P16, L11 to P17, L9. These results are interesting but a little out of place in a paper on "drought monitoring using thermal remote sensing", as you stated in L10 (A more detailed analysis is required...). Above all, this analysis suggests, again, how the adopted modeling framework may not be ideal for the study of this specific biome. Please justify this analysis in the context of the main goal of the study (thermal remote sensing), and against the use of ET/ETo as drought proxy.

---

## Referee Comment (RC3) · Anonymous Referee #3 · 15 Jul 2020

General comments:

The study by Gonzalez-Dugo et al. presents an interesting analysis of long-term ET and drought indicators over an Oak savanna region in Spain. The study implemented a surface energy balance model (i.e. SEBS) together with MODIS products and ERA meteorological data to obtain monthly and annual water stress indicators for a 17-year period. The manuscript demonstrated a sound remote sensing-based methodology and is valuable to better understand the long-term effects of droughts over an important and complex region such as the Spanish dehesa, which may be also relevant for other similar savanna-like ecosystems. The analysis of the monthly and annual time-series

demonstrated an important dataset that helps to better characterize and understand drought events (and their effects) in these water-limited ecosystems. The results and conclusions were well described and articulated.

However, I have some comments related to certain details of the model set-up, which were missing or not clearly elaborated in the methodology section. Since the study presents a workflow to obtain long-term water stress indicators, more information on how the input datasets were pre-processed is needed (e.g. retrievals of inputs, resampling of datasets at different temporal and spatial resolution) so this workflow can be reproduced for other studies/applications. Additionally, it was not very clear how the authors tackled the issue of having different vegetation covers (i.e. trees and grasses) and if the model inputs/structure reflected this added uncertainty in these types of landscapes. The retrieval of certain inputs, especially important ones like LAI and canopy height, should be more clearly described. In addition, the study should more clearly show the particularities of the dehesa system and how the methods presented here are more sound for monitoring dehesa (and similar) ecosystems compared to other ET products such as, for example, the MODIS ET product.

The study is concise and relatively well written. However, the authors should review certain sentences and try to write with more direct language in certain situations (see the specific comments below for examples).

Overall, I would recommend accepting this manuscript after revising and addressing the comments specified below.

Specific comments:

L44-45: Here, the authors briefly mention the complex canopy structure of the agrosystem and how it causes an added difficulty to assess and monitor droughts. However, a few more details on the particularities of dehesa/savanna ecosystems is needed in the introduction and, more concretely, why these ecosystems demonstrate greater uncertainty when using modeling methods, such as surface energy balance models,

especially compared to landscapes with more homogeneous canopy covers and structures. This would further justify the study, which provides a methodology that monitors ET and drought for an ecosystem that tends to be poorly represented by land-atmospheric models, usually causing for greater uncertainties.

L74: Why was SEBS used compared to other models? A small justification is needed for the use of SEBS. What advantages does it present compared to other models? Why not other thermal-based SEB models such as e.g. METRIC, SEBAL, TSEB etc or optical-based PM/PT methods as used in the MODIS ET product. Or even the use of products from geostationary satellites such as LSA-SAF ET.

L116: It says 'The green canopy cover and leaf area index (L) were calculated using the following equations (Choudhury et al., 1994)' however equation 8 or 9 do not detail how leaf area index was computed (only fractional cover, fc)

L125-129: It is not very clear how the canopy height was estimated. Is the canopy height assumed to be 8m, as such only accounting for tree and neglecting the grass/pasture or is it an integrated/effective value based on NDVI? If not ignoring the grass, how is the grass canopy height estimated? What is the relationship between NDVI and canopy height? I suggest to re-write this paragraph to makes this clearer and more specific.

L131-132: Leaf area index was previously defined as L in L116 but here uses the acronym LAI. Should be consistent throughout the manuscript.

L151-153: Review sentence with more direct language. E.g. 'The good correspondence between the model input was verified [. . .]'

Section 2: Some more clarification is needed in the methodology section on how the model inputs and parameters were set up and evaluated. Perhaps also a table that states all the inputs and parameters used in SEBS with their values/method would help clarify this. This information is scattered in the text but should be directly and clearly

stated in the methods. Were the input datasets filtered for cloud cover/quality? Looking at Table 1, the different datasets used have different temporal and spatial resolutions (additionally in the text it says MODIS LAI product was used but it is not shown in Table 1). So how were these datasets homogenized? Which resampling algorithm was used? Was everything averaged for the month? Was only daytime meteorological data used or also nighttime? All this information should be stated so that the presented method is reproducible. In addition, the model evaluation method, and criteria (e.g, RMSE, R2 etc) should be explicitly stated in this section.

L186: MBE acronym was not defined.

L202-204: review sentence 'A few of the years [..] an increase in run-off'

L218: Here it is mentioned that drought was evaluated at the annual scale but how was it aggregated? As an annual average or cumulative over the year?

L222-223: why is the drought event of 2016/2017 considered mild, if it reaches similar levels as the years 2004/2005 and 2011/12, which were considered the most severe droughts (Fig.4)? Is there a cutoff/threshold?

L225-228: Review sentence 'Figure 5 aggregates [..] scarcity on the system'. Sentence is too long, maybe cut in two with more direct language.

L263-264: Make sentence more direct 'The duration [. . .] these periods'.

Section 3.2: It would maybe be interesting to do a trend analysis to investigate if drought events are becoming more frequent/severe? Probably the time series is not large enough... but it does seem that the there are slightly more negative anomalies (particularly for Sta. Clo) from 2013/2014 onwards.

L293: More direct language, e.g. 'The SEBS model was used [..]'.

L317-19: Review sentence. More direct language, e.g. 'The approach proved useful [..] defining and identifying areas of interest for future studies at finer resolutions'.

Table 1: In table caption, it says from 2000-2015 but the study time period is 2001-2018 right?

Figure 6: The dehesa area of interest should be made more explicit and clearer in the map and legend. Also, little spatial analysis was provided in the text. For example, there seems to be important differences and patterns in the northern tip compared to the rest of the area of interest, most clearly seen in the average ET/ET0 map or in 2004/05, 2008/09, and 2011/12.

Figure 7a: There is no legend for the dashed green line.

All figures: There should be self-explanatory captions in all figures so that the reader can understand the figure without looking at the main text.

---

## Author Comment (AC1) · 20 Aug 2020

*Reviewer comments are typed in black colour, whereas the responses are typed in* blue *colour.*

General Comments The manuscript present am interesting study using a long-term dataset to characterize the impact of water stress on the dehesa region of Spain. Overall, study was well designed, the paper is well written, and the results and conclusions are fully supported. however there are a few aspects of the study that need some clarification. The concerns, along with handful of minor grammar and typographical errors, are noted below.

We thank the reviewer for the constructive comments. We have considered all of them, the suggested changes and clarifications are detailed here and will be introduced in the revised manuscript.

Specific Comments

1. Line 13: The sentence beginning "Drought is a ..." might be expressed more clearly as : "Drought is a devastating natural hazard that is difficult to define, detect and quantify."

The sentence will be changed

2. Line 13: The sentence beginning "Global meteorological data ..." is oddly constructed. It might be more clearly expressed as" The increased availability of both meteorological and remotely sensed data provides an opportunity to develop new methods to identify drought conditions and characterize how it changes over space and time."

The sentence will be changed

3. Line 26: The sentence beginning "During the drier ..." is unclear and needs revision.

The sentence will be changed to: "During the drier events, the changes in the grasslands and oak trees ground cover allowed a separate analysis of the strategies adopted by the two strata to cope with water stress".

4. Line 34: The sentence beginning "Drought is a ..." could be expressed more clearly if constructed as: "Drought, which is both a devastating natural hazard and globally widespread, has complex consequences across spatiotemporal scales and sectors."

The sentence will be changed to the proposed construction.

5. Line 43: Replace "slow-onset nature" with "slow onset".

It will be replaced

6. Line 48: Indicators of what?

Indicators of drought, it will be clarified in the manuscript.

7. Line 53: The sentence beginning "LST and VIs" reads oddly. The authors seem to be saying that by combining information about the surface temperature and vegetation, remote sensing-based models can provide accurate estimates of ET. But, rather than statin that explicitly, the coach it in terms of vegetation indices etc.

The sentence will be changed to 'LST and VIs have been used to provide ET estimations over agriculture ...

8. Line 115: This paragraph is a bit unclear. The authors state the parameterization of green vegetation fraction and height are unique for the dehesa. Are the authors back calculating the leaf area index (L) using equations 8 & 9? If so, why? Also, there is no discussion of canopy height and how it's calculation is modified to better represent the dehesa.

Yes, we obtained Fc using eq. 8 and L is derived from Fc using eq.9. To clarify the procedure, we will modify eq. 9 to provide a more direct computation of L.

The computation of the canopy height is described in the manuscript, but the paragraph is unclear, and it will be modified. Considering that the tree stratum of the dehesa is quite homogeneous in composition, dominated by mature Quercus ilex sp., and that grassland canopy has a very high variability of low height herbaceous species, the ecosystem structure has been simplified to compute hc in the following way: A constant height of 8 m has been assigned to oak trees, which is multiplied by its ground coverage in each pixel. Oaks fc is computed annually using summer NDVI in eq. 8. During the summer the grasslands are dry, and the only photosynthetically active vegetation contributing to the NDVI signal are the oak trees. The grassland height is low (< 1 m), affecting the effective canopy height of each pixel less than the trees, and it is also difficult to computed based on monthly vegetation indices given the high species variability. For this reason, the grassland height has been discarded and only the contribution of trees was considered to compute hc. We are aware that this is a simplification of a complex system that will contribute to the error of modelled fluxes. However, it was an operative solution considering the scale of this study.

9. Line 172: It would be helpful if the authors included a histogram and an estimate of the distribution skewness for ET and relative ET. From the description given here it appears quite small.

In the figure below (Fig1-commentR1), we present the histograms (one for each month) of both variables. For both variables most months presented an approximately symmetric distribution, with skewness between -0.5 and 0.5, three of them were moderately skewed and only one month (for ET) and two months (for ET/ETo) were slightly above one. We will elaborate this point in the manuscript. However, given the limited number of available points, these graphs only provide preliminary information and more data is required to confirm this point. For this reason, we prefer to include these graphs as supplementary information and not as part of the paper.

[Figure]

Fig1-commentsR1

10. Line 182: Replace "presented a general good agreement" with "generally showed good agreement".

It will be replaced

11. Line 184: Why the greater discrepancy for the turbulent fluxes compared to the C2 non-turbulent fluxes? Is this linked to imperfect closure for the flux measurements? Errors in partitioning the available energy between H and LE?

The imperfect energy balance closure is certainly a reason, as well as the discrepancy in the footprints of the different sensors (radiometer, soil heat flux plates, and the instruments for measuring the turbulent fluxes) and that of SEBS estimates. However, the different complexity in the formulation and computation of the radiative and the turbulent fluxes (the net radiation equation is a kind of linear representation, while the equation to estimate the sensible heat flux is highly non-linear), and the factors that influence each component (Rn is influenced by LWD, SWD, albedo and LST; H is influenced by LST, Ta, wind speed, NDVI, fc and LAI) also influence the final error. A small bias in LST, Ta, and vegetation information can cause a high bias in H (and thereby LE, compute as a residual). The soil heat flux has usually a low RMSD, but generally this comes with a higher relative error, due to the reduced magnitude of this flux.

12. Line 206: The sentence beginning "Very low runoff ..." is redundant and could be omitted.

It will be deleted

13. Line 207: Why isn't the relationship shown? Although it reasonable to suspect these two quantities would be correlated, a "close" relationship is a bit of a surprise. It would be useful to show this relational.

We show below (F2-commentsR1) the relationship between annual run-off measured at the Sta.Clo catchment reservoir and the annual aridity index (Budyko, 1974) estimated for the same catchment on the left, and the same relationship with the run-off (Q) also normalized by precipitation on the right. The shape of these relationships shows how variations in climate, as represented by variations of P and ETo, impact runoff and could provide a mean to assess the effects of a changing climate on water availability in this watershed. The budyko model represented in (b) was derived using Zhang et al. (2008) eq.9 with an adjusted value for $\alpha$ parameter equal to 0.54. It shows a mean to estimate long term annual run-off values in this catchment. Although these are interesting relationships, useful to complement the drought assessment, it's a little outside of the topic and might disrupt the flow of the results, so we prefer to present it as supplementary material.

[Figure]

F2-commentsR1. (a) Relationship between annual run-off measured at the Sta.Clo catchment reservoir and the annual aridity index (Budyko, 1974) estimated for the same catchment and (b) Relationship between run-off coefficient measured at the Sta.Clo catchment and the annual aridity index.

14. Line 207: Numerous metrics and indices have proposed been proposed over time to quantify aridity. It would be helpful to add a sentence or two to describe this index.

We will include in the text the following definition: "These annual run-off measurements followed a close relationship (Figure 4) with the annual aridity index (Budyko, 1974), estimated at Sta.Clo following Arora (2002), as the ratio between potential evaporation and annual precipitation."

15. Line 222: Do the difference in the anomalies suggest local drought conditions? For example, during 2008/2009 there is a strongly negative value at the ES-LMa site while the value is slightly positive at StaClo. Would this indicate a local drought in the area about ES-LMa?

This is a correct observation. The difference is caused by the big difference in precipitation, as indicated in Fig. 3, precipitation at Sta.Clo (683 mm/a) is about twice that at ES-LMa (338 mm/a).

16. Line 253: it worth point out that the peak in the autumn is much weaker than the one earlier in the year.

Yes, it will be pointed out in the revised manuscript.

17. Line 299: The phrase "and the more ..." also refers to ES-LMa, which was already discussed.

Yes, it will be deleted

18. Figure 5: The word "fraction" is misspelled.

It will be corrected

**References:**

Arora, V. K.: The use of the aridity index to assess climate change effect on annual runoff. J. Hidrol. 265:164-177. 2002.

Budyko, M.I.: Climate and life, Academic Press, Orlando, FL, 1974.

Zhang L., N. Potter, K. Hickel, Y. Zhang, Q. Shao:  Water balance modeling over variable time scales based on the Budyko framework – Model development and testing, J. Hydrol., 360: 117-131. https://doi.org/10.1016/j.jhydrol.2008.07.021. 2008

---

## Author Comment (AC2) · 20 Aug 2020

*Reviewer comments are typed in black colour, whereas the responses are typed in blue colour.*

General Comment

This paper deals with the modeling of drought in a oak savanna in Spain, where trees and pasture coexists, using ET estimates from thermal remote sensing data. I found the paper generally well written and well organized. The goal is clear, and the results sufficiently elaborate. However, I have three main concerns regarding the adopted methodology:

We really appreciate the time dedicated by the reviewer to read this manuscript and all the suggestions and comments that have been provided. We have considered all the comments, and the suggested changes and clarifications will be introduced in the revised manuscript.

1) the SEBS model is well-known in the remote sensing community for "instantaneous" application at the satellite overpass time (eventually followed by upscaling procedures to daily/monthly scale). Here the model is used on monthly data, but the authors fail to clarify how the model was adapted for the change in time scale (more details in the specific comment P8, L4).

The methodological section did not provide sufficient detail on how the different time step data were aggregated to SEBS inputs for the calculation. We will add a new section to the revised manuscript dealing with "Model parametrization and dataset preparation" to clarify this issue. The monthly ET calculation using SEBS was demonstrated by Chen et al. (2014). The structure of the model was not changed regardless of whether it was used for instantaneous, daily or monthly ET calculations. The difference in its implementation was only due to the input datasets. For monthly ET calculation, monthly mean LST, air temperature, wind speed, downward shortwave radiation, downward longwave radiation etc were used. The accuracy of monthly LST, a key variable in SEB models, was evaluated by Chen et al. 2017, supporting its applicability for climate studies and numerical model evaluation.

References:

Chen X, Z Su, Y Ma, J Cleverly, M Liddell. (2017) An accurate estimate of monthly mean land surface temperatures from MODIS clear-sky retrievals, , Journal of hydrometeorology 18 (10), 2827-2847

Chen X, Z Su, Y Ma, S Liu, Q Yu, Z Xu. (2014), Development of a 10-year (2001-2010) 0.1 data set of land-surface energy balance for mainland China. Atmos. Chem. Phys., 14, 13097–13117. 2014. www.atmos-chem-phys.net/14/13097/2014/

2) The authors decided to use anomalies of the ratio ET/ETo as drought indicator. However, they do not provided neither evidences that this index perform better than others (e.g. even the simple ET), nor justification on why this index was used for the ecosystem under analysis (is it better suited for oak savanna than others?). Indeed, part of the study shift the focus on fc, because ET is not able to separate the behavior of trees and pastures. This analysis, even if interesting, is out of place give the declared goal of the study.

The reasons for the use of evapotranspiration anomalies to assess agricultural drought and a remote sensing-based surface energy balance model to estimate ET are provided in the introduction. However, as the reviewer indicates, the selection of the ratio of actual to potential ET was not explained in the manuscript. The reason why ET is normalised by $ET_o$ is to separate the ET signal component responding to soil moisture from variations due to the radiation load. Therefore, this reduces the variability in ET due to seasonal variations in available energy. Anderson et al., (2011) showed that anomalies in $ET/ET_o$ were more strongly correlated with other drought indices (including the US Drought Monitor, PDSI, PDMI, PHDI, SPI) than were anomalies in ET for most US climatic divisions, showing strong agreements in the southwest of the country, with a similar climate to the study area.

However, following the reviewer's recommendation, a comparison between both series of anomalies (including also anomalies of Fc) has been performed (see figures below). The result showed that, for the conditions of the study, the anomalies of ET and $ET/ET_o$ performed similarly to characterize drought periods, presenting a high correlation ($R^2$=0.76 at monthly scale and $R^2$=0.82 at seasonal scale). It suggests that ET anomalies could be an option to monitor drought in dehesa areas. Nevertheless, the computation of $ET_o$ does not require additional variables than those already used by the energy balance models, with a quite straightforward computation. Once actual ET is estimated, the computation of $ET/ET_o$ takes very little effort and adds some confidence to the focus on the soil moisture signal. The graph of comparison of monthly anomalies will be added as a new figure to the paper and these results will be discussed in the text, including some comments to the rest of figures, presented as separate supplementary information. The justification of the selection of the ratio $ET/ET_o$ will be also included in the revised manuscript.

The explanation for the use of Fc and its connection to the goal of the paper is included in the following comment.

[Figure]

3) The authors used vegetation coverage and wheat productions as proxy of the drought impacts, without providing any justification for this choice. The first quantity is actually one of the input of

SEBS, but is also weirdly used also for "validation", whereas the second is not necessarily related to drought impacts in a drought-resistant agropastoral system (see their words in P3, L6 of the manuscript).

The vegetation condition and the failure of crops are known consequences of a declining soil moisture and both have been used previously as indicators of drought (Liu and Kogan, 1996; FAO, 1983). Both variables, together with general numbers of hydroelectricity production, were the only available data that can provide a complementary view on drought impact in addition to evapotranspiration anomalies. As the reviewer points out, the green canopy cover is one of the inputs of SEBS and it is not used in the manuscript to validate the series of ET/ETo anomalies. However, the explicit analysis of its evolution sheds some light on the interpretation of these anomalies. In the case of wheat production, this rainfed winter cereal is the main agricultural use of dehesa areas. It is periodically sown in many pasture fields of this ecosystem. Its growth cycle is similar to that of the natural grasslands, with both of them escaping drought and coping with the long summer dry season by completing its life cycle before serious soil and plant water deficits develop. Given that no irrigation is provided, the impact of moisture deficits over its yield can be considered an indirect indicator of the impact of drought on all dehesa herbaceous vegetation.

An explanation justifying the use of both proxies will be included in the methodological section of the manuscript.

References:

Liu, W.T., Kogan, F.N., 1996. Monitoring regional drought using the vegetation condition index. Int. J. Remote Sens. 17, 2761–2782.

Food and Agriculture Organization, 1983. Guidelines: Land evaluation for Rainfed Agriculture. FAO Soils Bulletin 52, Rome.

In view of these considerations, I suggest the authors to revisit the manuscript to clarify these points before considering for publication. Some additional specific comments are also reported below, which I hope would be useful for improving the overall quality of the manuscript.

Specific comments

Title: I would replace the world "monitoring" with something else, since in my opinion monitoring implies something done in near-real time.

We will replace the term *monitoring* by *assessment* in the title: "Long-term water stress and drought assessment of Mediterranean oak savanna vegetation using thermal remote sensing" and in some references along the text.

P2, L1: RMSD > xxx, and R2 < xxxx for all...

To provide a general idea of the global performance, we prefer to show average values rather than absolute ones for RMSD and R2. We will modify this sentence of the abstract to clarify it.

P2, L2-3: The details for each site are not needed in the abstract, especially after the previous sentence.

We are sorry but we are not sure to which "details for each site in the abstract" the reviewer's comment refers to. We don't provide details for each site separately there. There is a general comment "for both sites", which we consider relevant for the abstract.

P2, L8: "with the first one being. . .". I suggest to move this to a new sentence.

It will be changed.

P5, L4: Here I miss something that better links the previous description of the dehesa with the adopted modeling framework. In particular, why ET modeled by SEBS has been used? Is it a good option to capture the specificities of this environment (e.g. other options, such as dual source approaches, agri-forest modeling)?

We have not performed a comparison of different models' performance over this ecosystem. Several inter-comparison studies have evaluated different modelling schemes and no single one had been found consistently best across all biomes (Ershadi et al., 2013). The SEBS model has been selected here because it presents a good compromise between the detailed parameterization of the turbulent heat fluxes for different states of the land surface on the one hand, and the input requirements, kept to a feasible minimum and without requirements for local calibration, on the other. Thus, it is a good candidate to produce global fluxes (Chen et al. 2019, Timmermans et al., 2013) and this work may contribute to improve the model parametrization for this type of ecosystems, usually poorly represented in land-atmospheric models. There was another practical reason in that the model had been previously applied with good results by Chen et al., (2014), at a similar spatiotemporal scale. Many operative solutions presented in that paper were also used here, simplifying the implementation of the model.

References:

Chen X., Z. Su, Y. Ma: Remote sensing of global monthly evapotranspiration with an energy balance (EB) model. The International Archives of the Photogrammetry, Remote Sensing and Spatial Information Sciences, Volume XLII-2/W13, 2019 ISPRS Geospatial Week, Enschede, The Netherlands. https://doi.org/10.5194/isprs-archives-XLII-2-W13-1729-2019. 2019

Chen X, Z Su, Y Ma, S Liu, Q Yu, Z Xu. (2014), Development of a 10-year (2001-2010) 0.1 data set of land-surface energy balance for mainland China. Atmos. Chem. Phys., 14, 13097–13117. 2014. www.atmos-chem-phys.net/14/13097/2014/

Ershadi A., M.F. McCabe, J.P. Evans, N.W. Chaney, E.F. Wood: Multi-site evaluation of terrestrial evaporation models using FLUXNET data. Ag. Forest Meteorol. 187: 46–61 http://dx.doi.org/10.1016/j.agrformet.2013.11.008

Timmermans J., Z. Su, C. van der Tol, A. Verhoef, and W. Verhoef: Quantifying the uncertainty in estimates of surface–atmosphere fluxes through joint evaluation of the SEBS and SCOPE models.

Hydrol. Earth Syst. Sci., 17, 1561–1573, 2013. doi:10.5194/hess-17-1561-2013 www.hydrol-earth-syst-sci.net/17/1561/2013/

P6, L3: I would suggest to write the eq. as LE = Rn - . . .. since you already introduced concept of LE as residual.

It will be changed.

P6, Eqs. (4) and (5). The second eq. is redundant.

Eq.5 will be removed

P7, Rqs. (6) and (7). These two equations are confusing. In LEwet is computed via (6), then Hwet needs to be defined in another way, or vice versa. Please clarify.

Eq. (6) will be similarly removed, Eq. 16, in Su (2002), will be added for the calculation of H_wet.

Su Z.: The surface energy balance system (SEBS) for estimation of turbulent heat fluxes. Hydrol. Earth Sys. Sci., 6(1), 85– 99, 2002.

P7, L3. The way the limits are used needs a better clarification.

P7, L4. if Hwet is derived from Eq. (7), LEwet needs to be defined by an eq. that is not (6) (e.g. Penman-Monteith as stated afterward).

P7, L5. ". . . a set of assumptions. . .". Please provide a brief description of these assumptions.

Answer to the three questions above regarding the limits: The use of the limits in SEBS are fully described in Su (2002). We will rewrite this part of the SEBS model description to clarify it.

P7, L7. The role of canopy height is not clear at this point for a reader that is not familiar with the model. Please briefly introduce where and how hc plays a role. Also, the authors introduced a "revised version of the model. . . new bare soil resistance" (P5, L18), but the role of this new parameterization is not clear since there are no mention of resistance in the model description.

The canopy height is needed for calculating the momentum roughness length and thus, important for the sensible heat calculation. A short explanation will be added on how resistance is calculated, where the role of soil resistance appears.

P8, L4. The SEBS model has been designed for "instantaneous" application at the time of LST acquisition. As a consequence, more details needs to be provided on how the authors adapted the model to work on monthly LST. I think that the idea is to use monthly LST as a "artificial" instantaneous LST for a theoretical average day, but some questions that needs to be addressed are: - how did you ensure consistency between the mosaicked monthly LST and 6h ECMWF meteo forcing? - How 16 days NDVI was used jointly with monthly LST? - How daily upscale was performed? - How monthly upscale was performed?

This issue was mostly addressed in the first point of the general comments. It was clarified that no model upscale was performed. LST and all the meteo forcing used to run the SEBS model in this study were monthly mean values. Monthly mean meteo forcing were directly provided by ECMWF (available for download in its website). Monthly mean LST was processed following the work of Chen et al. (2017) referenced above. Monthly NDVI was derived from 16 days NDVI by selecting the maximum values in each month. All this information will be added to a new methodological section dealing with dataset preparation.

P9, L1. Some details on the balance closure would be helpful. Was closure forced, and with which method? How were the data cumulated at monthly scale (I'm assuming some unavoidable missing data during the acquisition, any constrain on minimum data, etc.)?

The closure of the balance was forced using the residual method. For ES_LMa the processing of the data (gap filling, monthly aggregation) corresponded to the procedure standardized by Fluxnet (described here: https://fluxnet.org/data/fluxnet2015-dataset/data-processing/). In the case of Sta.Clo, the comparison period was selected attending to the quality of the data and some month were discarded due to missing information. A new paragraph with the details on data selection and processing will be included in the manuscript.

P10, L13. It would be interesting to have a couple of words on the reason behind the use of ET/ETo rather than ET itself for the computation of anomalies. In my experience, there are many cases where ET anomalies are a better proxy of drought that ET/ETo ones. Ideally, the authors should add a test showing that ET/ETo outperform ET alone (especially with the latter being a more conservative approach, which does not need any additional quantity).

We will add to the revised version an explanation for the reasons behind the use of ET/ETo rather than ET itself for the computation of anomalies. In addition, we have compared both anomalies at monthly and seasonal scales, part of this analysis will be presented in the manuscript with a new figure and the rest as supplementary information.

P10, L15. I have some issue with the use of fc as proxy of drought impacts, especially when fc is also one of the input of SEBS. If fc is a good proxy of drought impact, why we should use a complex model such as SEBS (which uses fc as input) to derive a quantity (ET) which performance is evaluated against fc. Why don't we use directly fc (or rather fc anomalies) at this point?

We have justified above the way of using fc in the paper. Regarding the evaluation of Fc anomalies, a new analysis has been performed to compare its performance to drought assessment with ET and ET/ETo anomalies (see figures of the general comment 2). From these figures, both at monthly and seasonal scales, it can be derived that the drought events identified using the three variables would have been the same, but with different intensities and duration. The main differences can be found during the cold winter months when the vegetation is largely dormant. In these cases, the anomalies of Fc, similarly to the performance of other indices based on vegetation as the Vegetation Condition Index (VCI) (Heim, 2002) have a limited utility. The results are more comparable and could be more useful during the growing season.

Heim, R. R., 2002: A Review of Twentieth-Century Drought Indices Used in the United States. *Bull. Amer. Meteor. Soc.*, **83**, 1149–1166, https://doi.org/10.1175/1520-0477-83.8.1149.

P10, L18. Similarly, I miss the connection between the impact of drought on the dehesa (a predominantly oak savanna) and wheat production. I know that having an independent estimate of drought impacts is tricky, but if the focus of the paper is specifically for the dehesa, you should justify better why wheat production is a good proxy of the drought impact on a likely drought-resistant, adapted oak savanna. The use of this quantity risks to lost the specificity of the work that you introduced earlier.

We have justified the use of wheat production as a component of dehesa and attending to its similar growth cycle to natural grasslands. The impact of moisture deficits over its yield can be considered an indirect indicator of the impact of drought on dehesa herbaceous vegetation. This point will be clarified in the methodology of the revised manuscript.

P11, L12. It would be better to have the results disaggregated for seasons, in order to better highlight the impact of this seasonality in the error. This would help discussing the results, since drought may be mostly concentrate in some seasons. Also, since your goal is to use ET/ETo anomalies as proxy for drought, it would be much better to have in addition a validation of both ET/ETo values and z values against ground data. Even if the length of the time series is quite short, it is important to show that the model is able to capture the year-to-year fluctuations, since this is what you want to reproduce. Often, ET estimates are "well" modeled only because the area has a strong yearly cycle.

Of the different temporal scales to show the results, we have selected the most extreme available (year and month). The seasonal information can be derived from Figure 7 for ET, ET0, P and fc and the identified dry period. The validation of ET estimated, as shown in Figure 2, is performed on monthly data.

P12, L1. It is weird to me that you show the yearly-aggregated data before the monthly one. Apart form that, Figure 3 is a good example of my consideration on P10, L13. Just looking at the plot, it seems that ET capture the same events that ETo if Precipitation is used as reference. What is the added value of using ET/ETo rather than ET alone?

We chose to present the results from a coarser temporal scale to provide a more general vision of the evolution of drought years to more detailed monthly results in which we can discuss shorter term variations.

P12, l15 to P13, L5. This whole paragraph seems a little out of topic to me. I suggest to reword to clarify the role in explaining drought in the region, or remove it completely.

This first part of paragraph (in our text lines 207-212) is intended to describe the area of study in terms of aridity and provide some numbers corresponding to the experimental sites, to classify them in relation with other climate areas of the world. Information regarding this analysis will be included as supplementary material, and some clarifications will be added to the text. The second part (lines 212-216) compares the two sites and discusses some aspects of Figure 3, as the relation between ET and ETo at annual scale, that we consider related to the topic of the paper.

P13, L13. Please define a mild drought. Also, it is not clear to me what is the role of this intercomparison between the modeled data over the two sites. Please clarify the aim of this comparison and justify the inclusion of a dedicated figure.

We define drought intensity in terms of maximum negative anomaly of relative ET values reached during the event (thus using the standard deviation as a measure of its departure from the mean). For the analysis of the events that occurred during the study period, the following thresholds were used: severe drought (anomalies <=-1.5); moderate drought (anomalies between -1 and -1.5) and mild drought (anomalies between -1 and 0). These classes are used for both annual and monthly time steps. This info will be added.

The intercomparison between sites complements the information provided on the experimental sites used to validate the model. In addition, we don't present a complete disaggregate analysis and most of the paper is focused on the whole dehesa region. This figure of the experimental sites points out that the general patterns are similar but there exist local differences and provides an estimate of the magnitude of these differences.

Fig. 5. Again, what is the added value of ET/ETo anomalies over ET alone (or, even worse, fc)? If anything, these figures are convincing me even more that a complex modeling framework is not needed, at least at annual scale. I'm sure that there is something more, but this is not discussed and justified by the accompanying text.

This issue is addressed above and also in the manuscript, including a new analysis and a new figure comparing the anomalies of ET/ETo, ET and fc at monthly scale. In the complementary information the analysis is extended to the seasonal scale.

Fig. 6. There is an odd strikingly resemblance between the spatial patterns in the years 2004/2005 and 2011/2012. Can you elaborate a little more on that?

Yes, both maps look quite similar, but they are different. An option for the analysis could be to produce a difference map to analyze similarities and differences.

P15, L2. Is this the average over the whole dehesa? A single point? Other? Please clarify. Also, in case of the average, it would be interesting to see if also the spatial variability (std.dev) shows interesting results.

Yes, it is the average. We will calculate the std.dev and will include it in the paper if it provides interesting results.

P15, L12. What about the intra-annual fluctuactions? Are they similar to ET/ETo z values also at this temporal scale? Any temporal delay?

We don't fully understand this question, we presented the monthly data to analyze the intra-annual fluctuations. The comment has a different number of page/line than the manuscript we have. In most comments, we have identified the reference attending to the content but in this case it's not completely clear.

P15, L19. Similarly to comment P13, L13, duration and intensity of drought needs to be defined in the methodology section.

We define the duration of the drought as the successive number of months with negative anomalies and the intensity as the maximum anomaly in this continuous period. These definitions will be clarified in the methodology.

P16, L11 to P17, L9. These results are interesting but a little out of place in a paper on "drought monitoring using thermal remote sensing", as you stated in L10 (A more detailed analysis is required...). Above all, this analysis suggests, again, how the adopted modeling framework may not be ideal for the study of this specific biome. Please justify this analysis in the context of the main goal of the study (thermal remote sensing), and against the use of ET/ETo as drought proxy.

The focus of the paper is on the assessment of long-term water stress and drought in dehesa ecosystem, the means used is thermal remote sensing, and fc evolution is also used to interpret anomalies of relative ET. The modelling framework used here is not the only plausible approach to monitor drought in this biome. However, the results have shown that it is well fitted for this system.

---

## Author Comment (AC3) · 20 Aug 2020

*Reviewer comments are typed in black colour, whereas the responses are typed in* blue *colour.*

General comments:

The study by Gonzalez-Dugo et al. presents an interesting analysis of long-term ET and drought indicators over an Oak savanna region in Spain. The study implemented a surface energy balance model (i.e. SEBS) together with MODIS products and ERA meteorological data to obtain monthly and annual water stress indicators for a 17-year period. The manuscript demonstrated a sound remote sensing-based methodology and is valuable to better understand the long-term effects of droughts over an important and complex region such as the Spanish dehesa, which may be also relevant for other similar savanna-like ecosystems. The analysis of the monthly and annual time-series demonstrated an important dataset that helps to better characterize and understand drought events (and their effects) in these water-limited ecosystems. The results and conclusions were well described and articulated.

However, I have some comments related to certain details of the model set-up, which were missing or not clearly elaborated in the methodology section. Since the study presents a workflow to obtain long-term water stress indicators, more information on how the input datasets were pre-processed is needed (e.g. retrievals of inputs, resam- pling of datasets at different temporal and spatial resolution) so this workflow can be reproduced for other studies/applications. Additionally, it was not very clear how the authors tackled the issue of having different vegetation covers (i.e. trees and grasses) and if the model inputs/structure reflected this added uncertainty in these types of land-scapes. The retrieval of certain inputs, especially important ones like LAI and canopy height, should be more clearly described. In addition, the study should more clearly show the particularities of the dehesa system and how the methods presented here are more sound for monitoring dehesa (and similar) ecosystems compared to other ET products such as, for example, the MODIS ET product.

The study is concise and relatively well written. However, the authors should review certain sentences and try to write with more direct language in certain situations (see the specific comments below for examples).

Overall, I would recommend accepting this manuscript after revising and addressing the comments specified below.

We really appreciate the time dedicated by the reviewer to read this manuscript and all the suggestions and comments that have been provided. We have considered all the comments, and the suggested changes and clarifications will be introduced in the revised manuscript.

Specific comments:

L44-45: Here, the authors briefly mention the complex canopy structure of the agro-system and how it causes an added difficulty to assess and monitor droughts. However, a few more details on the particularities of dehesa/savanna ecosystems is needed in the introduction and, more concretely, why these ecosystems demonstrate greater uncertainty when using modeling methods, such as surface energy balance models especially compared to landscapes with more homogeneous canopy covers and structures. This would further justify the study, which provides a methodology that monitors ET and drought for an ecosystem that tends to be poorly represented by land-atmospheric models, usually causing for greater uncertainties.

Similarly to other savanna ecosystems, the different components of dehesa structure: sparse tall vegetation, large areas of grasses, shrubs, and bare soil, contribute differently to the turbulent exchange and radiative transfer, hindering its modeling, especially when compared with more homogeneous landscapes. In addition, these vegetation layers differ in phenology, physiology and function: while the trees are evergreen and have access to sources of water all year, the herbaceous layer only taps water from the first cm of soil and dries up during summer. The combined different functioning and characteristics of the system components affects the exchange of sensible and latent heat flux, resulting in a high spatial and temporal flux variability difficult to account for in model parametrization and algorithms. This structure appears to play an important role in savannas' resilience, making the system an efficient convector of sensible heat and keeping the canopy surface temperature inside the adequate range for survival (Baldocchi et al., 2004). A brief explanation of this will be added to the introduction of the paper.

References:

Baldocchi, Dennis D. and Xu, Liukang and Kiang, Nancy. (2004) How plant functional-type, weather, seasonal drought, and soil physical properties alter water and energy fluxes of an oak-grass savanna and an annual grassland Agricultural and Forest Meteorology, 123: 13-39. doi: https://doi.org/10.1016/j.agrformet.2003.11.006

L74: Why was SEBS used compared to other models? A small justification is needed for the use of SEBS. What advantages does it present compared to other models? Why not other thermal-based SEB models such as e.g. METRIC, SEBAL, TSEB etc or optical-based PM/PT methods as used in the MODIS ET product. Or even the use of products from geostationary satellites such as LSA-SAF ET.

We have not performed a comparison of different models' performance over this ecosystem. Several inter-comparison studies have evaluated different modelling schemes and no single one has been found consistently best across all biomes (Ershadi et al., 2013). The SEBS model has been selected here because it presents a good compromise between the detailed parameterization of the turbulent heat fluxes for different states of the land surface on the one hand, and the input requirements, kept to a feasible minimum and without requirements for local calibration, on the other. Thus, it is a good candidate to produce global fluxes (Chen et al. 2019, Timmermans et al., 2013) and this work may contribute to improve the model parametrization for this type of ecosystems, usually poorly represented in land-atmospheric models as the reviewer mentioned in the previous point. There was another practical reason, in that the model had been previously applied, with good results by Chen et al., (2014), at a similar spatiotemporal scale that the one of

interest for this application. Many operative solutions presented in that paper were used also here, simplifying the implementation of the model.

References:

Chen X., Z. Su, Y. Ma: Remote sensing of global monthly evapotranspiration with an energy balance (EB) model. The International Archives of the Photogrammetry, Remote Sensing and Spatial Information Sciences, Volume XLII-2/W13, 2019 ISPRS Geospatial Week, Enschede, The Netherlands. https://doi.org/10.5194/isprs-archives-XLII-2-W13-1729-2019. 2019

Chen X, Z Su, Y Ma, S Liu, Q Yu, Z Xu. (2014), Development of a 10-year (2001-2010) 0.1 data set of land-surface energy balance for mainland China. Atmos. Chem. Phys., 14, 13097–13117. 2014. www.atmos-chem-phys.net/14/13097/2014/

Ershadi A., M.F. McCabe, J.P. Evans, N.W. Chaney, E.F. Wood: Multi-site evaluation of terrestrial evaporation models using FLUXNET data. Ag. Forest Meteorol. 187: 46–61 http://dx.doi.org/10.1016/j.agrformet.2013.11.008

Timmermans J., Z. Su, C. van der Tol, A. Verhoef, and W. Verhoef: Quantifying the uncertainty in estimates of surface–atmosphere fluxes through joint evaluation of the SEBS and SCOPE models. Hydrol. Earth Syst. Sci., 17, 1561–1573, 2013. doi:10.5194/hess-17-1561-2013 www.hydrol-earth-syst-sci.net/17/1561/2013/

L116: It says 'The green canopy cover and leaf area index (L) were calculated using the following equations (Choudhury et al., 1994)' however equation 8 or 9 do not detail how leaf area index was computed (only fractional cover, fc)

Fc is calculated using eq.8 and L is derived from fc using eq. 9. However, to clarify the procedure we will modify eq. 9 to provide a more direct computation of L.

L125-129: It is not very clear how the canopy height was estimated. Is the canopy height assumed to be 8m, as such only accounting for tree and neglecting the grass/pasture or is it an integrated/effective value based on NDVI? If not ignoring the grass, how is the grass canopy height estimated? What is the relationship between NDVI and canopy height? I suggest to re-write this paragraph to makes this clearer and more specific.

Yes, we will rewrite the paragraph to clarify computation of canopy height and justify the decisions made to simplify the structure of the system. This simplification is based on the homogeneity in composition of the tree stratum of the dehesa, dominated by mature Quercus ilex sp., and on the very high variability of herbaceous species with low heights of the grassland canopy. To compute hc a constant height of 8 m has been assigned to oak trees, which is multiplied by its ground coverage in each pixel. The oaks fc is computed annually using summer NDVI in eq.8. During the summer the grasslands are dry, and the only photosynthetically active vegetation contributing to the NDVI signal are the oak trees. The grassland height is low (< 1 m), affecting the effective canopy height of each pixel less than that of the trees, and it is also difficult to compute based on monthly vegetation indices given the high species variability. For this reason, the grassland height has been discarded and only the contribution of trees was considered to compute hc. We are aware that this is a simplification of a complex system that will contribute to the error of modelled fluxes. However, it was an operative solution considering the scale of this study.

L131-132: Leaf area index was previously defined as L in L116 but here uses the acronym LAI. Should be consistent throughout the manuscript.

Yes, it will be corrected

L151-153: Review sentence with more direct language. E.g. 'The good correspondence between the model input was verified [. . .]'

It will be changed to: "In both cases, the good correspondence between the model input and the ground measurements was verified (data not shown)."

Section 2: Some more clarification is needed in the methodology section on how the model inputs and parameters were set up and evaluated. Perhaps also a table that states all the inputs and parameters used in SEBS with their values/method would help clarify this. This information is scattered in the text but should be directly and clearly stated in the methods. Were the input datasets filtered for cloud cover/quality? Looking at Table 1, the different datasets used have different temporal and spatial resolutions (additionally in the text it says MODIS LAI product was used but it is not shown in Table 1). So how were these datasets homogenized? Which resampling algorithm was used? Was everything averaged for the month? Was only daytime meteorological data used or also nighttime? All this information should be stated so that the presented method is reproducible. In addition, the model evaluation method, and criteria (e.g, RMSE, R2 etc) should be explicitly stated in this section.

The methodological section 2.1 will be reformulated to include the missing information about the application of the model described in the referee's comment. It will be divided in three subsections: 2.1.1 SEBS model description; 2.1.2 Model parametrization and dataset preparation; and 2.1.3. Model evaluation. 2.1.2 will include a new version of table 1, a detailed description of the parameters, and the datasets used in the model, including the explanation of datasets resampling and homogenization.

L186: MBE acronym was not defined.

MBE stands for Mean Bias Error; it will be defined in the text.

L202-204: review sentence 'A few of the years [..] an increase in run-off'

It will be changed to: "Very wet years, and those with average rainfall but intense precipitation events producing an increase in run-off, did not follow this pattern."

L218: Here it is mentioned that drought was evaluated at the annual scale but how was it aggregated? As an annual average or cumulative over the year?

The annual value was an average of monthly anomalies. We will add this information to the text.

L222-223: why is the drought event of 2016/2017 considered mild, if it reaches similar levels as the years 2004/2005 and 2011/12, which were considered the most severe droughts (Fig.4)? Is there a cutoff/threshold?

Yes, we will define drought intensity in terms of maximum negative anomaly of relative ET values reached during the event (thus using the standard deviation as a measure of its departure from the mean). When analyzing the events occurred during the study period, the following thresholds were used: severe drought (anomalies <=-1.5); moderate drought (anomalies between -1 and -1.5) and mild drought (anomalies between -1 and 0). These classes are used for both annual and monthly time steps. In terms of intensity, only the drought event of 2004/2005 can be considered severe (max negative anomaly = -1.7) and 2016/2017 is classified as moderate with the maximum negative anomaly equal to -1.29.

L225-228: Review sentence 'Figure 5 aggregates [..] scarcity on the system'. Sentence is too long, maybe cut in two with more direct language.

The sentence will be changed to: "Figure 5 aggregates, for the total dehesa area, the evolution of the relative ET anomalies, together with the exchanges of energy between the surface and the atmosphere, the green canopy cover, and the production of rainfed wheat. The last two variables were selected as indicators of the impact of water scarcity on the system."

L263-264: Make sentence more direct 'The duration [. . .] these periods'.

This sentence will be changed to use more direct language but also to include a definition of drought intensity attending to a previous comment.

Section 3.2: It would maybe be interesting to do a trend analysis to investigate if drought events are becoming more frequent/severe? Probably the time series is not large enough... but it does seem that the there are slightly more negative anomalies (particularly for Sta. Clo) from 2013/2014 onwards.

This is an interesting analysis that we would like to perform when a longer dataset is available. The current database, as the reviewer mentioned, is not large enough and it could provide misleading information.

L293: More direct language, e.g. 'The SEBS model was used [..]'.

The sentence of L293 will be changed and the whole text will be reviewed to use a more direct language.

L317-19: Review sentence. More direct language, e.g. 'The approach proved useful [..] defining and identifying areas of interest for future studies at finer resolutions'.

The sentence will be changed to: The approach proved useful for providing insight into the characteristics of drought events over this ecosystem, and for defining and identifying areas of interest for future studies at finer resolutions.

Table 1: In table caption, it says from 2000-2015 but the study time period is 2001-2018 right?

Yes, it was a mistake, it's 2001-2018 and will be corrected in the manuscript

Figure 6: The dehesa area of interest should be made more explicit and clearer in the map and legend. Also, little spatial analysis was provided in the text. For example, there seems to be important differences and patterns in the northern tip compared to the rest of the area of interest, most clearly seen in the average ET/ET0 map or in 2004/05, 2008/09, and 2011/12.

We will modify Figure 6 to make the area of interest more explicit in the map and the legend and we will add a few sentences briefly dealing with the spatial analysis. However, we have not performed a detailed analysis and only general comments can be made.

Figure 7a: There is no legend for the dashed green line.

The explanation has been added to the caption of Figure 7.

All figures: There should be self-explanatory captions in all figures so that the reader can understand the figure without looking at the main text.

The figure captions will be corrected to:

Figure 1: Distribution of oak savanna area in the Iberian Peninsula. Location of Sta.Clo (Santa Clotilde) and ES-LMa (Las Majadas) validation sites and pictures of both eddy covariance flux towers.

Figure 2. Comparison of monthly energy fluxes of latent heat (LE), sensible heat (H), net radiation (Rn) and soil heat flux (G) estimated using the SEBS model at a monthly scale and observed fluxes at each oak savanna site: ES-LMa (LA) for the years 2009-2011 and Sta.Clo (SC) for the years 2015-2017.

Figure 3. Evolution of annual rainfall, ET, ETo and ET/ETo at ES-LMa site (a) and Sta.Clo site (b), and annual run-off at Sta.Clo watershed from the hydrological years 2001/02 to 2017/2018.

Figure 4. Annual anomalies of relative evapotranspiration at ES-LMa and Sta.Clo experimental sites estimated using the SEBS model from 2001/02 to 2017/18.

Figure 5. Evolution from 2001/02 to 2017/18 of annual anomalies of relative evapotranspiration, energy balance components, air and surface temperature, vegetation ground fraction cover and rainfed wheat yield, aggregated for the whole oak savanna area of the Iberian Peninsula.

Figure 6. Spatial distribution of annual anomalies of relative evapotranspiration for the oak savanna area of the Iberian Peninsula from 2001/02 to 2017/18, the average ET/ETo for the period and its standard deviation (STD)

Figure 7. (a) Monthly evolution of evapotranspiration anomalies (blue line), with negative values indicating drier than normal conditions (depicted in red), and green canopy cover (green line) of the oak savanna area of the Iberian Peninsula. The dashed green lines connect the annual maximum

and minimum values of fc; (b) Monthly evolution of and rainfall, ETo and ET in the same region and time interval.

---

## Author Response (AR1)

*Reviewer comments are typed in black colour, whereas the responses are typed in* blue *colour.*

General Comments The manuscript present am interesting study using a long-term dataset to characterize the impact of water stress on the dehesa region of Spain. Overall, study was well designed, the paper is well written, and the results and conclusions are fully supported. however there are a few aspects of the study that need some clarification. The concerns, along with handful of minor grammar and typographical errors, are noted below.

We thank the reviewer for the constructive comments. We have considered all of them, the suggested changes and clarifications are detailed here and have been introduced in the revised manuscript.

Specific Comments

1. Line 13: The sentence beginning "Drought is a ..." might be expressed more clearly as : "Drought is a devastating natural hazard that is difficult to define, detect and quantify."

Sentence changed (line 13).

2. Line 13: The sentence beginning "Global meteorological data ..." is oddly constructed. It might be more clearly expressed as" The increased availability of both meteorological and remotely sensed data provides an opportunity to develop new methods to identify drought conditions and characterize how it changes over space and time."

The sentence has been changed (line 13-15).

3. Line 26: The sentence beginning "During the drier ..." is unclear and needs revision.

The sentence has been changed to: "During the drier events, the changes in the grasslands and oak trees ground cover allowed a separate analysis of the strategies adopted by the two strata to cope with water stress". (line 27).

4. Line 34: The sentence beginning "Drought is a ..." could be expressed more clearly if constructed as: "Drought, which is both a devastating natural hazard and globally widespread, has complex consequences across spatiotemporal scales and sectors."

The sentence has been changed to the proposed construction (line 36).

5. Line 43: Replace "slow-onset nature" with "slow onset".

6. Line 48: Indicators of what?

7. Line 53: The sentence beginning "LST and VIs" reads oddly. The authors seem to be saying that by combining information about the surface temperature and vegetation, remote sensing-based models can provide accurate estimates of ET. But, rather than statin that explicitly, the coach it in terms of vegetation indices etc.

The sentence has been deleted and simplified to: 'SEBM have been used to provide ET estimations over agriculture … and agroforestry systems. (line 55).

8. Line 115: This paragraph is a bit unclear. The authors state the parameterization of green vegetation fraction and height are unique for the dehesa. Are the authors back calculating the leaf area index (L) using equations 8 & 9? If so, why? Also, there is no discussion of canopy height and how it's calculation is modified to better represent the dehesa.

Yes, we obtained Fc using eq.  8 and L is derived from Fc using eq.9. To clarify the procedure, we have modified eq. 8 and 9 to provide a more direct computation of L.

The computation of the canopy height was described in the manuscript, but the paragraph was unclear, and it has been modified (lines 155-165). Considering that the  tree stratum of the dehesa is quite homogeneous in composition, dominated by mature Quercus ilex sp., and that grassland canopy has a very high variability of low height herbaceous species, the ecosystem structure has been simplified to compute hc in the following way: A constant height of 8 m has been assigned to oak trees, which is multiplied by its ground coverage in each pixel. Oaks fc is computed annually using summer NDVI in eq. 8. During the summer the grasslands are dry, and the only photosynthetically active vegetation contributing to the NDVI signal are the oak trees. The grassland height is low (< 1 m), affecting the effective canopy height of each pixel less than the trees, and it is also difficult to computed based on monthly vegetation indices given the high species variability. For this reason, the grassland height has been discarded and only the contribution of trees was considered to compute hc. We are aware that this is a simplification of a complex system that will contribute to the error of modelled fluxes. However, it was an operative solution considering the scale of this study.

9. Line 172: It would be helpful if the authors included a histogram and an estimate of the distribution skewness for ET and relative ET. From the description given here it appears quite small.

In the Figure S1 (supplement), we present the histograms (one for each month) of both variables. For both variables most months presented an approximately symmetric distribution, with skewness between -0.5 and 0.5, three of them were moderately skewed and only one month (for ET) and two months (for ET/ETo) were slightly above one. We have elaborated this point in the manuscript (lines 236-242). However, given the limited number of available points, these graphs only provide preliminary information and more data is required to confirm this point. For this

reason, we prefer to include these graphs as supplementary information and not as part of the paper.

10. Line 182: Replace "presented a general good agreement" with "generally showed good agreement".

Replaced (line 260).

11. Line 184: Why the greater discrepancy for the turbulent fluxes compared to the C2 non-turbulent fluxes? Is this linked to imperfect closure for the flux measurements? Errors in partitioning the available energy between H and LE?

The imperfect energy balance closure is certainly a reason, as well as the discrepancy in the footprints of the different sensors (radiometer, soil heat flux plates, and the instruments for measuring the turbulent fluxes) and that of SEBS estimates. However, the different complexity in the formulation and computation of the radiative and the turbulent fluxes (the net radiation equation is a kind of linear representation, while the equation to estimate the sensible heat flux is highly non-linear), and the factors that influence each component (Rn is influenced by LWD, SWD, albedo and LST; H is influenced by LST, Ta, wind speed, NDVI, fc and LAI) also influence the final error. A small bias in LST, Ta, and vegetation information can cause a high bias in H (and thereby LE, compute as a residual). The soil heat flux has usually a low RMSD, but generally this comes with a higher relative error, due to the reduced magnitude of this flux.

12. Line 206: The sentence beginning "Very low runoff ..." is redundant and could be omitted.

The sentence has been deleted (line 285).

13. Line 207: Why isn't the relationship shown? Although it reasonable to suspect these two quantities would be correlated, a "close" relationship is a bit of a surprise. It would be useful to show this relational.

In the supplement (Figure S2.) is shown the relationship between annual run-off measured at the Sta.Clo catchment reservoir and the annual aridity index (Budyko, 1974) estimated for the same catchment on the left, and the same relationship with the run-off (Q) also normalized by precipitation on the right. The shape of these relationships showed how variations in climate, as represented by variations of P and ETo, impact runoff and could provide a mean to assess the effects of a changing climate on water availability in this watershed. The budyko model represented in (b) was derived using Zhang et al. (2008) eq.9 with an adjusted value for α parameter equal to 0.54. It shows a mean to estimate long term annual run-off values in this catchment. Although these are interesting relationships, useful to complement the drought assessment, it's a little outside of the topic and might disrupt the flow of the results, so we prefer to present it as supplementary material.

14. Line 207: Numerous metrics and indices have proposed been proposed over time to quantify aridity. It would be helpful to add a sentence or two to describe this index.

We have included in the text the following definition: "Annual……. the annual aridity index (Budyko, 1974) estimated at Sta.Clo following Arora (2002), as the ratio between potential evaporation and annual precipitation." (line 288-289).

15. Line 222: Do the difference in the anomalies suggest local drought conditions? For example, during 2008/2009 there is a strongly negative value at the ES-LMa site while the value is slightly positive at StaClo. Would this indicate a local drought in the area about ES-LMa?

This is a correct observation. The difference is caused by the big difference in precipitation, as indicated in Fig. 3, precipitation at Sta.Clo (683 mm/a) is about twice that at ES-LMa (338 mm/a).

16. Line 253: it worth point out that the peak in the autumn is much weaker than the one earlier in the year.

Yes, it has been pointed out in the revised manuscript (lines 334,335).

17. Line 299: The phrase "and the more ..." also refers to ES-LMa, which was already discussed.

It has been deleted (line 392)

18. Figure 5: The word "fraction" is misspelled.

Corrected

However, following the reviewer's recommendation, a comparison between both series of anomalies (including also anomalies of Fc) has been performed (see figures below, in the supplement and new Fig 8). The result showed that, for the conditions of the study, the anomalies of ET and $ET/ET_o$ performed similarly to characterize drought periods, presenting a high correlation ($R^2$=0.76 at monthly scale and $R^2$=0.82 at seasonal scale). It suggests that ET anomalies could be an option to monitor drought in dehesa areas. Nevertheless, the computation of $ET_o$ does not require additional variables than those already used by the energy balance models, with a quite straightforward computation. Once actual ET is estimated, the computation of $ET/ET_o$ takes very little effort and adds some confidence to the focus on the soil moisture signal. The graph of comparison of monthly anomalies has been included as the new figure 8 in the paper and these results are now discussed in the text (lines 373-383), including some comments to the rest of figures, presented as separate supplementary information. The justification of the selection of the ratio $ET/ET_o$ is also included in the revised manuscript.

The explanation for the use of fc and its connection to the goal of the paper is included in the following comment.

[Figure]

Figure comment 2. Comparison of ET/ETo, ET and *fc* anomalies at monthly and seasonal scale.

[Figure]

Figure S4. Relationships of $ET/ET_0$ and ET anomalies at monthly and seasonal scales (left figures) and ET/ETo and ET anomalies at monthly and seasonal scales (right figures).

3) The authors used vegetation coverage and wheat productions as proxy of the drought impacts, without providing any justification for this choice. The first quantity is actually one of the input of SEBS, but is also weirdly used also for "validation", whereas the second is not necessarily related to drought impacts in a drought-resistant agropastoral system (see their words in P3, L6 of the manuscript).

The vegetation condition and the failure of crops are known consequences of a declining soil moisture and both have been used previously as indicators of drought (Liu and Kogan, 1996; FAO, 1983). Both variables, together with general numbers of hydroelectricity production, were the only available data that can provide a complementary view on drought impact in addition to evapotranspiration anomalies. As the reviewer points out, the green canopy cover is one of the inputs of SEBS and it is not used in the manuscript to validate the series of ET/ETo anomalies. However, the explicit analysis of its evolution sheds some light on the interpretation of these anomalies. In the case of wheat production, this rainfed winter cereal is the main agricultural use of dehesa areas. It is periodically sown in many pasture fields of this ecosystem. Its growth cycle is similar to that of the natural grasslands, with both of them escaping drought and coping with the long summer dry season by completing its life cycle before serious soil and plant water deficits develop. Given that no irrigation is provided, the impact of moisture deficits over its yield can be considered an indirect indicator of the impact of drought on all dehesa herbaceous vegetation.

An explanation justifying the use of both proxies has been be included in the methodological section of the manuscript (lines 248-254)

References:

Liu, W.T., Kogan, F.N., 1996. Monitoring regional drought using the vegetation condition index. Int. J. Remote Sens. 17, 2761–2782.

Food and Agriculture Organization, 1983. Guidelines: Land evaluation for Rainfed Agriculture. FAO Soils Bulletin 52, Rome.

In view of these considerations, I suggest the authors to revisit the manuscript to clarify these points before considering for publication. Some additional specific comments are also reported below, which I hope would be useful for improving the overall quality of the manuscript.

Specific comments

Title: I would replace the world "monitoring" with something else, since in my opinion monitoring implies something done in near-real time.

We have replaced the term *monitoring* by *assessment* in the title: "Long-term water stress and drought assessment of Mediterranean oak savanna vegetation using thermal remote sensing" and in some references along the text.

P2, L1: RMSD > xxx, and R2 < xxxx for all...

To provide a general idea of the global performance, we prefer to show average values rather than absolute ones for RMSD and R2. We have modified this sentence of the abstract to clarify it (line 20).

P2, L2-3: The details for each site are not needed in the abstract, especially after the previous sentence.

We are sorry but we are not sure to which "details for each site in the abstract" the reviewer's comment refers to. We don't provide details for each site separately there. There is a general comment "for both sites", which we consider relevant for the abstract.

P2, L8: "with the first one being. . .". I suggest to move this to a new sentence.

It has been changed (line 25).

P5, L4: Here I miss something that better links the previous description of the dehesa with the adopted modeling framework. In particular, why ET modeled by SEBS has been used? Is it a good option to capture the specificities of this environment (e.g. other options, such as dual source approaches, agri-forest modeling)?

We have not performed a comparison of different models' performance over this ecosystem. Several inter-comparison studies have evaluated different modelling schemes and no single one has been found consistently best across all biomes (Ershadi et al., 2013). The SEBS model has been selected here because it presents a good compromise between the detailed parameterization of the turbulent heat fluxes for different states of the land surface on the one hand, and the input requirements, kept to a feasible minimum and without requirements for local calibration, on the other (explanation added to the introduction, lines 58-60). Thus, it is a good candidate to produce global fluxes (Chen et al. 2019, Timmermans et al., 2013) and this work may contribute to improve the model parametrization for this type of ecosystems, usually poorly represented in land-atmospheric models. There was another practical reason, which is that the model had been previously applied with good results by Chen et al., (2014), at a similar spatiotemporal scale. Many operative solutions presented in that paper were also used here, simplifying its implementation.

P7, L7. The role of canopy height is not clear at this point for a reader that is not familiar with the model. Please briefly introduce where and how hc plays a role. Also, the authors introduced a "revised version of the model. . . new bare soil resistance" (P5, L18), but the role of this new parameterization is not clear since there are no mention of resistance in the model description.

The canopy height is needed for calculating the momentum roughness length and thus, important for the sensible heat calculation. This information has been added to the text (lines 155-165). Concerning the bare soil resistance, as the reviewer observed, it is not mentioned in this brief

description of the model. To avoid confusion, we have deleted this comment and we refer the reader interested in a complete description of the model to previous papers (lines 103- 104).

P8, L4. The SEBS model has been designed for "instantaneous" application at the time of LST acquisition. As a consequence, more details needs to be provided on how the authors adapted the model to work on monthly LST. I think that the idea is to use monthly LST as a "artificial" instantaneous LST for a theoretical average day, but some questions that needs to be addressed are: - how did you ensure consistency between the mosaicked monthly LST and 6h ECMWF meteo forcing? - How 16 days NDVI was used jointly with monthly LST? - How daily upscale was performed? - How monthly upscale was performed?

This issue was mostly addressed in the first point of the general comments. It was clarified that no model upscale was performed. LST and all the meteo forcing used to run the SEBS model in this study were monthly mean values. Monthly mean meteo forcing were directly provided by ECMWF (available for download in its website). Monthly mean LST was processed following the work of Chen et al. (2017) referenced above. Monthly NDVI was derived from 16 days NDVI by selecting the maximum values in each month. All this information has been added to a new methodological section (2.2) dealing with dataset preparation.

P9, L1. Some details on the balance closure would be helpful. Was closure forced, and with which method? How were the data cumulated at monthly scale (I'm assuming some unavoidable missing data during the acquisition, any constrain on minimum data, etc.)?

The closure of the balance was forced using the residual method. For ES_LMa the processing of the data (gap filling, monthly aggregation) corresponded to the procedure standardized by Fluxnet (described here: https://fluxnet.org/data/fluxnet2015-dataset/data-processing/). In the case of Sta.Clo, the comparison period was selected attending to the quality of the data and some month were discarded due to missing information. A new paragraph with some details on data selection and processing has been added to the manuscript. (lines 199-203).

P10, L13. It would be interesting to have a couple of words on the reason behind the use of ET/ETo rather than ET itself for the computation of anomalies. In my experience, there are many cases where ET anomalies are a better proxy of drought that ET/ETo ones. Ideally, the authors should add a test showing that ET/ETo outperform ET alone (especially with the latter being a more conservative approach, which does not need any additional quantity).

We have included in the revised version of the manuscript an explanation for the reasons behind the use of ET/ETo rather than ET itself for the computation of anomalies. In addition, we have compared both anomalies at monthly and seasonal scales, part of this analysis will be presented in the manuscript with a new figure and the rest as supplementary information. Detailed information regarding this point has been previously described.

P10, L15. I have some issue with the use of fc as proxy of drought impacts, especially when fc is also one of the input of SEBS. If fc is a good proxy of drought impact, why we should use a complex model such as SEBS (which uses fc as input) to derive a quantity (ET) which performance is evaluated against fc. Why don't we use directly fc (or rather fc anomalies) at this point?

We have justified above the way of using fc in the paper. Regarding the evaluation of Fc anomalies, a new analysis has been performed to compare its performance to drought assessment with ET and

ET/ETo anomalies (see figures of the general comment 2). From these figures, both at monthly and seasonal scales, it can be derived that the drought events identified using the three variables would have been the same, but with different intensities and duration. The main differences can be found during the cold winter months when the vegetation is largely dormant. In these cases, the anomalies of Fc, similarly to the performance of other indices based on vegetation as the Vegetation Condition Index (VCI) (Heim, 2002) have a limited utility. The results are more comparable and could be more useful during the growing season. This discussion has been added to the text. However, the explicit analysis of fc evolution sheds some light on the interpretation of these anomalies. An explanation justifying the use of this proxy has been included in the methodological section of the manuscript (lines 248-254).

Heim, R. R., 2002: A Review of Twentieth-Century Drought Indices Used in the United States. *Bull. Amer. Meteor. Soc.*, **83**, 1149–1166, https://doi.org/10.1175/1520-0477-83.8.1149.

P10, L18. Similarly, I miss the connection between the impact of drought on the dehesa (a predominantly oak savanna) and wheat production. I know that having an independent estimate of drought impacts is tricky, but if the focus of the paper is specifically for the dehesa, you should justify better why wheat production is a good proxy of the drought impact on a likely drought-resistant, adapted oak savanna. The use of this quantity risks to lost the specificity of the work that you introduced earlier.

We have justified the use of wheat production as a component of dehesa and attending to its similar growth cycle to natural grasslands. The impact of moisture deficits over its yield can be considered an indirect indicator of the impact of drought on dehesa herbaceous vegetation. This point has been clarified in the methodology of the revised manuscript (lines 252-254).

P11, L12. It would be better to have the results disaggregated for seasons, in order to better highlight the impact of this seasonality in the error. This would help discussing the results, since drought may be mostly concentrate in some seasons. Also, since your goal is to use ET/ETo anomalies as proxy for drought, it would be much better to have in addition a validation of both ET/ETo values and z values against ground data. Even if the length of the time series is quite short, it is important to show that the model is able to capture the year-to-year fluctuations, since this is what you want to reproduce. Often, ET estimates are "well" modeled only because the area has a strong yearly cycle.

Of the different temporal scales to show the results, we have selected the most extreme available (year and month). The seasonal information can be derived from Figure 7 for ET, ETo, P and fc and the identified dry period. The validation of ET estimated, as shown in Figure 2, is performed on monthly data.

P12, L1. It is weird to me that you show the yearly-aggregated data before the monthly one. Apart form that, Figure 3 is a good example of my consideration on P10, L13. Just looking at the plot, it seems that ET capture the same events that ETo if Precipitation is used as reference. What is the added value of using ET/ETo rather than ET alone?

We chose to present the results from a coarser temporal scale to provide a more general vision of the evolution of drought years to more detailed monthly results in which we can discuss shorter term variations.

P12, l15 to P13, L5. This whole paragraph seems a little out of topic to me. I suggest to reword to clarify the role in explaining drought in the region, or remove it completely.

This first part of paragraph (in the original text, lines 207-212) is intended to describe the area of study in terms of aridity and provide some numbers corresponding to the experimental sites, to classify them in relation with other climate areas of the world. Information complementing this description has been included as supplementary material (Figure S2), and some clarifications have been added to the text (lines 287- 289). The second part (in the original text, lines 212-216) compares the two sites and discusses some aspects of Figure 3, as the relation between ET and ETo at annual scale, that we consider related to the topic of the paper.

P13, L13. Please define a mild drought. Also, it is not clear to me what is the role of this intercomparison between the modeled data over the two sites. Please clarify the aim of this comparison and justify the inclusion of a dedicated figure.

We define drought intensity in terms of maximum negative anomaly of relative ET values reached during the event (thus using the standard deviation as a measure of its departure from the mean). For the analysis of the events that occurred during the study period, the following thresholds were used: severe drought (anomalies <=-1.5); moderate drought (anomalies between -1 and -1.5) and mild drought (anomalies between -1 and 0). These classes are used for both annual and monthly time steps. This information has been included in the revised text (lines 243-247).

The intercomparison between sites complements the information provided on the experimental sites used to validate the model. In addition, we don't present a complete disaggregate analysis and most of the paper is focused on the whole dehesa region. This figure of the experimental sites points out that the general patterns are similar but there exist local differences and provides an estimate of the magnitude of these differences.

Fig. 5. Again, what is the added value of ET/ETo anomalies over ET alone (or, even worse, fc)? If anything, these figures are convincing me even more that a complex modeling framework is not needed, at least at annual scale. I'm sure that there is something more, but this is not discussed and justified by the accompanying text.

This issue was addressed above and also in the manuscript, including a new analysis and a new figure comparing the anomalies of ET/ETo, ET and fc at monthly scale. In the complementary information the analysis is extended to the seasonal scale.

Fig. 6. There is an odd strikingly resemblance between the spatial patterns in the years 2004/2005 and 2011/2012. Can you elaborate a little more on that?

Yes, both maps look quite similar, but they are different. An option for the analysis could be to produce a difference map to analyze similarities and differences. This will be considered in a future analysis at local scale.

P15, L2. Is this the average over the whole dehesa? A single point? Other? Please clarify. Also, in case of the average, it would be interesting to see if also the spatial variability (std.dev) shows interesting results.

Yes, it is the average. We will analyze the std.dev in later work.

P15, L12. What about the intra-annual fluctuactions? Are they similar to ET/ETo z values also at this temporal scale? Any temporal delay?

We don't fully understand this question, we presented the monthly data to analyze the intra-annual fluctuations. The comment has a different number of page/line than the manuscript we have. In most comments, we have identified the reference attending to the content but in this case it's not completely clear.

P15, L19. Similarly to comment P13, L13, duration and intensity of drought needs to be defined in the methodology section.

We define the duration of the drought as the successive number of months with negative anomalies and the intensity as the maximum anomaly in this continuous period. These variables have been defined in the methodology.

P16, L11 to P17, L9. These results are interesting but a little out of place in a paper on "drought monitoring using thermal remote sensing", as you stated in L10 (A more detailed analysis is required...). Above all, this analysis suggests, again, how the adopted modeling framework may not be ideal for the study of this specific biome. Please justify this analysis in the context of the main goal of the study (thermal remote sensing), and against the use of ET/ETo as drought proxy.

The focus of the paper is on the assessment of long-term water stress and drought in dehesa ecosystem, the means used is thermal remote sensing, and *fc* evolution is also used to interpret anomalies of relative ET. The modelling framework used here is not the only plausible approach to monitor drought in this biome. However, the results have shown that it is well fitted for this system.

*Reviewer comments are typed in black colour, whereas the responses are typed in* blue *colour.*

General comments:

The study by Gonzalez-Dugo et al. presents an interesting analysis of long-term ET and drought indicators over an Oak savanna region in Spain. The study implemented a surface energy balance model (i.e. SEBS) together with MODIS products and ERA meteorological data to obtain monthly and annual water stress indicators for a 17-year period. The manuscript demonstrated a sound remote sensing-based methodology and is valuable to better understand the long-term effects of droughts over an important and complex region such as the Spanish dehesa, which may be also relevant for other similar savanna-like ecosystems. The analysis of the monthly and annual time-series demonstrated an important dataset that helps to better characterize and understand drought events (and their effects) in these water-limited ecosystems. The results and conclusions were well described and articulated.

However, I have some comments related to certain details of the model set-up, which were missing or not clearly elaborated in the methodology section. Since the study presents a workflow to obtain long-term water stress indicators, more information on how the input datasets were pre-processed is needed (e.g. retrievals of inputs, resam- pling of datasets at different temporal and spatial resolution) so this workflow can be reproduced for other studies/applications. Additionally, it was not very clear how the authors tackled the issue of having different vegetation covers (i.e. trees and grasses) and if the model inputs/structure reflected this added uncertainty in these types of land-scapes. The retrieval of certain inputs, especially important ones like LAI and canopy height, should be more clearly described. In addition, the study should more clearly show the particularities of the dehesa system and how the methods presented here are more sound for monitoring dehesa (and similar) ecosystems compared to other ET products such as, for example, the MODIS ET product.

The study is concise and relatively well written. However, the authors should review certain sentences and try to write with more direct language in certain situations (see the specific comments below for examples).

Overall, I would recommend accepting this manuscript after revising and addressing the comments specified below.

We really appreciate the time dedicated by the reviewer to read this manuscript and all the suggestions and comments that have been provided. We have considered and answered below all the comments. The suggested changes and clarifications have been introduced in the revised manuscript.

Specific comments:

L44-45: Here, the authors briefly mention the complex canopy structure of the agro-system and how it causes an added difficulty to assess and monitor droughts. However, a few more details on the particularities of dehesa/savanna ecosystems is needed in the introduction and, more concretely, why these ecosystems demonstrate greater uncertainty when using modeling methods, such as surface energy balance models especially compared to landscapes with more homogeneous canopy covers and structures. This would further justify the study, which provides a methodology that monitors ET and drought for an ecosystem that tends to be poorly represented by land-atmospheric models, usually causing for greater uncertainties.

Similarly to other savanna ecosystems, the different components of dehesa structure: sparse tall vegetation, large areas of grasses, shrubs, and bare soil, contribute differently to the turbulent exchange and radiative transfer, hindering its modeling especially when compared with more homogeneous landscapes. In addition, these vegetation layers differ in phenology, physiology and function: while the trees are evergreen and have access to deep sources of water all year, the herbaceous layer only taps water from the first cm of soil and dries up during summer. The combined different functioning and characteristics of the system components affects the exchange of sensible and latent heat flux, resulting in a high spatial and temporal flux variability difficult to account for in model parametrization and algorithms. This structure appears to play an important role in savannas' resilience, making the system an efficient convector of sensible heat and keeping the canopy surface temperature inside the adequate range for survival (Baldocchi et al., 2004). A brief explanation of the influence of dehesa characteristics on energy flux modelling has been added to the introduction of the paper (lines 79-87).

*Fc* is calculated using eq.8 and L is derived from *fc* using eq. 9. However, to clarify the procedure we have modified eq. 8 and 9 (new eq. 7 and 8) to provide a more direct computation of both variables.

L125-129: It is not very clear how the canopy height was estimated. Is the canopy height assumed to be 8m, as such only accounting for tree and neglecting the grass/pasture or is it an integrated/effective value based on NDVI? If not ignoring the grass, how is the grass canopy height estimated? What is the relationship between NDVI and canopy height? I suggest to re-write this paragraph to makes this clearer and more specific.

Yes, we have rewritten the paragraph to clarify computation of canopy height and justify the decisions made to simplify the structure of the system (lines 155-165). This simplification is based on the homogeneity in composition of the tree stratum of the dehesa, dominated by mature Quercus ilex sp., and on the very high variability of herbaceous species with low heights of the grassland canopy. To compute hc a constant height of 8 m has been assigned to oak trees, which is multiplied by its ground coverage in each pixel. The oaks fc is computed annually using summer NDVI in eq.8. During the summer the grasslands are dry, and the only photosynthetically active vegetation contributing to the NDVI signal are the oak trees. The grassland height is low (< 1 m), affecting the effective canopy height of each pixel less than that of the trees, and it is also difficult to compute based on monthly vegetation indices given the high species variability. For this reason, the grassland height has been discarded and only the contribution of trees was considered to compute

hc. We are aware that this is a simplification of a complex system that will contribute to the error of modelled fluxes. However, it was an operative solution considering the scale of this study.

L131-132: Leaf area index was previously defined as L in L116 but here uses the acronym LAI. Should be consistent throughout the manuscript.

Yes, it has been corrected (line 203)

L151-153: Review sentence with more direct language. E.g. 'The good correspondence between the model input was verified [. . .]'

It  be changed to: "In both cases, the good correspondence between the model input and the ground measurements was verified (data not shown)." (lines 208-209).

Section 2: Some more clarification is needed in the methodology section on how the model inputs and parameters were set up and evaluated. Perhaps also a table that states all the inputs and parameters used in SEBS with their values/method would help clarify this. This information is scattered in the text but should be directly and clearly stated in the methods. Were the input datasets filtered for cloud cover/quality? Looking at Table 1, the different datasets used have different temporal and spatial resolutions (additionally in the text it says MODIS LAI product was used but it is not shown in Table 1). So how were these datasets homogenized? Which resampling algorithm was used? Was everything averaged for the month? Was only daytime meteorological data used or also nighttime? All this information should be stated so that the presented method is reproducible. In addition, the model evaluation method, and criteria (e.g, RMSE, R2 etc) should be explicitly stated in this section.

The methodological section 2 has been reformulated to include the missing information about the application of the model described in the referee's comment. A new subsection "2.2 Model parametrization and dataset preparation" include all the missing information about the obtention and processing of model inputs and parameters. We have completed table 1 to account for all inputs, including LAI and *fc*, which were derived from MODIS NDVI, and no specific MODIS product were used for these variables. This is clearly detailed now in the new Table 1. It is also described in 2.2 section the use of monthly data, and that all datasets were spatially averaged or subdivided to a common resolution of 0.05°.  In addition, information about the model evaluation and definition of criteria has been added to the end of the validation section (lines 214-216), renamed to "2.3 Validation sites and model evaluation".

L186: MBE acronym was not defined.

MBE stands for Mean Bias Error and it is defined now in the text, (line 215).

L202-204: review sentence 'A few of the years [..] an increase in run-off'

It will be changed to: "Very wet years, and those with average rainfall but intense precipitation events producing an increase in run-off, did not follow this pattern." (line 280-281).

L218: Here it is mentioned that drought was evaluated at the annual scale but how was it aggregated? As an annual average or cumulative over the year?

The annual value was an average of monthly anomalies. This information was added to the revised version in lines 241-242.

L222-223: why is the drought event of 2016/2017 considered mild, if it reaches similar levels as the years 2004/2005 and 2011/12, which were considered the most severe droughts (Fig.4)? Is there a cutoff/threshold?

Yes, we have defined drought intensity in terms of the maximum negative anomaly of relative ET values reached during the event (thus using the standard deviation as a measure of its departure from the mean). When analyzing the events occurred during the study period, the following thresholds were used: severe drought (anomalies <=-1.5); moderate drought (anomalies between -1 and -1.5) and mild drought (anomalies between -1 and 0). These classes are used for both annual and monthly time steps. These definitions have been included in the description of the methodology (lines 245-247). In terms of intensity, only the drought event of 2004/2005 can be considered severe (max negative anomaly = -1.7) and 2016/2017 is classified as moderate with the maximum negative anomaly equal to -1.29.

L225-228: Review sentence 'Figure 5 aggregates [..] scarcity on the system'. Sentence is too long, maybe cut in two with more direct language.

The sentence will be changed to: "Figure 5 aggregates, for the total dehesa area, the evolution of the relative ET anomalies, together with the exchanges of energy between the surface and the atmosphere, the green canopy cover, and the production of rainfed wheat. The last two variables were selected as indicators of the impact of water scarcity on the system." Lines (305-309)

L263-264: Make sentence more direct 'The duration [. . .] these periods'.

As drought intensity and duration have been defined in the methodology (see a previous answer), the sentence has been shortened, lines 344-345.

Section 3.2: It would maybe be interesting to do a trend analysis to investigate if drought events are becoming more frequent/severe? Probably the time series is not large enough... but it does seem that the there are slightly more negative anomalies (particularly for Sta. Clo) from 2013/2014 onwards.

This is an interesting analysis that we would like to perform when a longer dataset is available. The current database, as the reviewer mentioned, is not large enough and it could provide misleading information.

L293: More direct language, e.g. 'The SEBS model was used [..]'.

The sentence of L293 has been changed (line386).

L317-19: Review sentence. More direct language, e.g. 'The approach proved useful [..] defining and identifying areas of interest for future studies at finer resolutions'.

The sentence is changed to: The approach proved useful for providing insight into the characteristics of drought events over this ecosystem, and for defining and identifying areas of interest for future studies at finer resolutions, (lines 410-412).

Table 1: In table caption, it says from 2000-2015 but the study time period is 2001-2018 right?

Yes, it was a mistake, it's 2001-2018 and has been corrected in the manuscript.

Figure 6: The dehesa area of interest should be made more explicit and clearer in the map and legend. Also, little spatial analysis was provided in the text. For example, there seems to be important differences and patterns in the northern tip compared to the rest of the area of interest, most clearly seen in the average ET/ET0 map or in 2004/05, 2008/09, and 2011/12.

We have modified Figure 6 to highlight the dehesa area in the figure and the legend. Regarding the spatial analysis, the comment in the text is indeed very reduced, since we have not performed a detailed analysis yet. Some differences can be observed visually but without a more careful analysis, we preferred to include only a general comment related to the experimental site's areas where we have better information.

Figure 7a: There is no legend for the dashed green line.

The explanation has been added to the caption of Figure 7 (line 608)

All figures: There should be self-explanatory captions in all figures so that the reader can understand the figure without looking at the main text.

The figure captions have been reviewed (lines 594-612)

[revised manuscript text omitted]

**modified Figure  5**

[Figure]

**modified Figure 6**

[Figure]

**Figure 7**

[Figure]

**new Figure 8**

---

## Author Response (AR2)

We really appreciate the time dedicated by both reviewers to read this manuscript and all the suggestions and comments that have been provided. We have considered and answered below all the comments. The suggested changes and clarifications have been introduced in the revised manuscript. Reviewer comments are typed in black colour, whereas the responses are typed in blue colour.

**Referee#1. Report 2**

General Comments
The manuscript presents a long-term analysis to characterize the impact of water stress on the dehesa region of Spain. Overall, study was well designed, the paper is well written, and the results and conclusions are fully supported. Additionally, the authors provided thoughtful responses to the comments regarding the earlier iteration of the manuscript. Thus, it is recommended that the paper should be accepted once the authors have addressed the handful of additional and relatively minor comments listed below.

Specific Comments
1. Line 43: The sentence beginning "However, the slow ..." could be better expressed as "However, the slow onset of drought, the large extension of savanna areas, and their complex canopy structure introduce additional difficulties to the challenge of monitoring drought and assessing its adverse effects."

The sentence has been changed

2. Line 55: The sentence beginning "In particular, the ..." could be expressed more clearly as "In particular, the SEBS (Surface Energy Balance System) model (Su, 2002) presents a good compromise between the detailed parameterization the turbulent heat fluxes for different states of the land surface and minimizing the input requirements of the model without the need for local calibration. "

The sentence has been changed

3. Line 69: The sentence beginning "Mediterranean oaks have …" could read more simply as "Mediterranean oaks can minimize the effects of water scarcity through a combination of physiological mechanisms that occur over a range of time scales Rambal (1993)."

4. Line 150: How were the input changed for monthly flux estimates?

The oaks fc is computed annually, and a single hc value was used for every month of a year. This is clarified now in the text.

5. Line 187: The hyphens in "root mean square error" are unnecessary and should be omitted. Similarly, the should be deleted in the "mean bias error' below.

Corrected

6. Line 197: Replace "than" with "as".

Replaced

7. Line 232: The RMSE for this study are modest and likely falls within the expected uncertainty of the measurements themselves. Also, since the model forces closure of the, most of the error in the turbulent fluxes can be attributed to the propagation of errors in the model estimates of Rn and G. This could be worth noting.

A sentence highlighting this observation has been added to the revised text.

8. Line 264: What were these hypotheses?

It is referred to the complementary hypothesis, as it is now clear in the text

**Referee#3. Report 2**

The authors have made the appropriate changes considering the comments made from the first round. The manuscript is clearer regarding the methods, model set-up and retrieval of inputs for the model application and analysis.

However, I recommend to revise one last time the writing style and semantics of the manuscript. In certain sections, the semantics can be improved by using clearer and direct language. Also, the manuscript should be revised generally to avoid the use of long-winded sentences (i.e. sentences that are > 4-5 lines). I therefore recommend to accept the manuscript for publication after revising some of the writing style. I leave below some specific recommendations:

We have revised the writing style of the whole text and rewrite the longest sentences.

L42 – Consider revising the sentence in L42-43. E.g. 'More recently, insurance services provide farmers with a means of recovery for pasture production negatively affected by disasters due to water stress.'

The sentence has been modified

L80 – change 'cm' to 'centimeters'

Changed

L292-93 – Consider revising the sentence in L292-293. E.g. 'The recovery of the vegetation water status, in most areas, was generally achieved the year following dry ones'
The sentence has been modified

L299-301 – Consider revising sentence to make it shorter (or break it into 2 sentences) and a comma (,) is needed after '[…] arrive' in L300

The sentence has been divided